# Is High Variance Unavoidable in RL? A Case Study in Continuous Control

**Johan Bjorck, Carla P. Gomes, Kilian Q. Weinberger**
Cornell University

## Abstract

Reinforcement learning (RL) experiments have notoriously high variance, and minor details can have disproportionately large effects on measured outcomes. This is problematic for creating reproducible research and also serves as an obstacle when applying RL to sensitive real-world applications. In this paper, we investigate causes for this perceived instability. To allow for an in-depth analysis, we focus on a specifically popular setup with high variance – continuous control from pixels with an actor-critic agent. In this setting, we demonstrate that poor outlier runs which completely fail to learn are an important source of variance, but that weight initialization and initial exploration are not at fault. We show that one cause for these outliers is unstable network parametrization which leads to saturating nonlinearities. We investigate several fixes to this issue and find that simply normalizing penultimate features is surprisingly effective. For sparse tasks, we also find that partially disabling clipped double Q-learning decreases variance. By combining fixes we significantly decrease variances, lowering the average standard deviation across 21 tasks by a factor $> 3$ for a state-of-the-art agent. This demonstrates that the perceived variance is not necessarily inherent to RL. Instead, it may be addressed via simple modifications and we argue that developing low-variance agents is an important goal for the RL community.

## 1 Introduction

With the advent of deep learning, reinforcement learning (RL) with function approximators has become a promising strategy for learning complex behavior (Silver et al., 2017; Mnih et al., 2015). While RL can work remarkably well, deep RL experiments have been observed to exhibit high variance (Chan et al., 2020; Clary et al., 2018; Lynnerup et al., 2020). Specifically, the rewards obtained by RL agents commonly vary significantly depending on the random seeds used for training, even if no other modifications are made. Furthermore, implementation level details (Engstrom et al., 2019) and even hardware scheduling (Henderson et al., 2018) often have an outsized effect on performance.

High variance is problematic for creating reproducible RL research, and achieving statistically significant results over large RL benchmarks is often prohibitively time-consuming (Colas et al., 2018; Agarwal et al., 2021). Furthermore, high variance provides an obstacle for real-world applications of RL, where safety and predictability can be critical (Garcıa & Fernández, 2015; Srinivasan et al., 2020). Especially relevant RL applications include medical testing (Bastani et al., 2021), autonomous vehicles (Bellemare et al., 2020) and communications infrastructure (Liu et al., 2020).

In this paper, we investigate causes for high variance in RL. To limit the paper's scope and computational footprint we focus on continuous control from pixels with a state-of-the-art (SOTA) actor-critic algorithm (Lillicrap et al., 2021; Yarats et al., 2021b). This is a popular setup that exhibits high variance, and where actor-critic algorithms often achieve SOTA results (Laskin et al., 2020b;a; Kostrikov et al., 2020). In this setting, we first show that an important cause of variance is outlier runs that fail early and never recover. It is natural to hypothesize that the typical sources of randomness, network initialization, and initial exploration, are to blame. However, we demonstrate that these do in fact have little impact on the final performance. We similarly provide experiments that suggest that neither poor feature representations nor excessive sensitivity of the environment is at fault. We instead demonstrate that unstable network parametrization can drive the variance and lead to excessively large activations that saturate action distributions. We propose several methods to

avoid such issues and find that they decrease variance. The most effective method is simply setting penultimate features to unit norm – a strategy we call *penultimate normalization* – which avoids exploding activations by design. We also find that partially disabling double clipped Q-learning (Fujimoto et al., 2018), a strategy we call *asymmetrically double clipped Q-learning*, reduces variance for many sparse tasks. By combining multiple approaches we increase average rewards while decreasing variance as measured of 21 tasks. The average standard deviation decreases by a factor $> 3$ and for some tasks by more than an order of magnitude. Our experiments suggest that high variance is not necessarily inherent to RL. Instead, one can directly optimize for lower variance without hurting the average reward. We argue that developing such low-variance agents is an important goal for the RL community – both for the sake of reproducible research and real-world deployment.

## 2 BACKGROUND

We consider continuous control modeled as a Markov decision process (MDP) with a continuous action space $\mathcal{A}$ and state space $\mathcal{S}$. In practice, the action space $\mathcal{A}$ is often bounded to account for physical limitations, e.g., how much force robotic joints can exert. A popular choice is simply $\mathcal{A} = [-1, 1]^{dim(\mathcal{A})}$. Within this setting, Deep Deterministic Policy Gradients (DDPG) (Lillicrap et al., 2021) is a performant algorithm that forms the backbone of many state-of-the-art agents (Laskin et al., 2020b;a; Kostrikov et al., 2020; Yarats et al., 2021b). DDPG uses a neural network (the *critic*) to predict the Q-value $Q_\phi(\mathbf{a}_t, \mathbf{s}_t)$ for each state-action pair. The parameters $\phi$ of the critic are trained by minimizing the soft Bellman residual:

$$\mathbb{E}\big(Q_\phi(\mathbf{a}_t, \mathbf{s}_t) - \big[r_t + \gamma \mathbb{E}[\hat{Q}(\mathbf{a}_{t+1}, \mathbf{s}_{t+1})]\big]\big)^2 \tag{1}$$

Here, $r_t$ is the reward and $\mathbb{E}[\hat{Q}(\mathbf{a}_{t+1}, \mathbf{s}_{t+1})]$ is the Q-value estimated by the *target* critic – a network whose weights can e.g. be the exponentially averaged weights of the critic. One can use multi-step returns for the critic target (Sutton, 1988). Another common trick is double clipped q-learning, i.e. using two Q-networks $Q_1, Q_2$ and taking $Q = \min(Q_1, Q_2)$ to decrease Q-value overestimation (Fujimoto et al., 2018). The policy is obtained via an *actor*-network with parameters $\theta$, which maps each state $\mathbf{s}_t$ to a distribution $\pi_\theta(\mathbf{s}_t)$ over actions. The action is obtained by calculating $\mathbf{a}_\theta^{pre}(\mathbf{s})$, the vector output of the actor network. Thereafter a $\tanh$ non-linearity is applied elementwise to ensure $\mathbf{a} \in [-1, 1]^n$ for $n = dim(\mathcal{A})$. The policy is then obtained as:

$$\mathbf{a}_\theta(\mathbf{s}) = \tanh(\mathbf{a}_\theta^{pre}(\mathbf{s})) + \boldsymbol{\epsilon} \tag{2}$$

$\boldsymbol{\epsilon}$ is some additive noise which encourages exploration, e.g. a Gaussian variable with time-dependent variance $\sigma_t^2$. Various noise types have been employed (Achiam, 2018; Lillicrap et al., 2021; Yarats et al., 2021b). Since the Q-values correspond to the expected discounted rewards, the actions $\mathbf{a}_\theta$ should maximize $Q_\phi(\mathbf{a}_\theta, \mathbf{s})$. Thus, to obtain a gradient update for the actor one uses the gradient $\nabla_\theta Q_\phi(\mathbf{a}_\theta, \mathbf{s})$, effectively calculating gradients *through* the critic network. In control from pixels, the state $\mathbf{s}$ is typically an image (possibly data augmented) processed by a convolutional network.

## 3 BENCHMARKING VARIANCE

**Experimental Setup.** To limit the paper's scope and computational footprint, we focus on continuous control from pixels with an actor-critic algorithm. This setup is relatively close to real-world robotics, where reliable performance is important. We consider the standard continuous control benchmark deepmind control (dm-control) (Tassa et al., 2020). Within dm-control, a long line of state-of-the-art models relies on the actor-critic setup (Laskin et al., 2020b;a; Kostrikov et al., 2020). We consider the recent data-augmentation based DDPG-type agent DRQv2 (Yarats et al., 2021b) which provides state-of-the-art performance and runs fast. We use the default hyperparameters that Yarats et al. (2021b) uses on the *medium* benchmark (listed in Appendix A) throughout the paper. We will refer to this agent as the *baseline* agent. We are interested in using a large number of random seeds to obtain accurate statistics and thus need to limit the number of environments we consider. To select environments we simply rank the dm-control tasks based upon their relative variance as measured by Yarats et al. (2021b). We select the five tasks with the highest relative variance – finger turn hard, walker walk, hopper hop, acrobot swingup, and reacher hard (see Table 4 in Appendix A for details). In Section 4.2 we consider a larger set of tasks. We will refer to these as *environments* or *tasks* interchangeably. For each run, we train the agent for one million frames, or equivalently 1,000

episodes, and evaluate over ten episodes. We use 10 or 40 seeds for each method we consider, these are not shared between experiments, but we reuse the runs for the baseline agent across figures.

## 3.1 VERIFYING VARIANCE

We first aim to verify that the environments indeed exhibit high variance. To this end, we run the baseline agent with 40 random seeds on each task. Figure 1 shows the result of this experiment for three tasks, see Appendix A for the two other tasks. We show both individual runs as well as mean and standard deviations. For some tasks, the variance is dominated by *outlier runs* that fail to learn early in the training and never recover. E.g. for the walker task, we see several runs which have not improved during the first 500 episodes, and most of these do not improve in the following 500 episodes either. Except for these outliers, the variance of the walker task is relatively small. We also note that these outliers separate themselves relatively early on, suggesting that there are some issues in early training that cause variance. If it was possible to eliminate poor outlier runs, we would expect the variance to go down. Motivated by these observations, and the walker task in particular, we will investigate various potential causes of outlier runs and high variance.

## 3.2 DOES INITIALIZATION MATTER?

There are multiple sources of randomness specific to the early training iterations. The agent networks are randomly initialized and the agent is given some initial experience, collected by random actions, to start training on. Could these two initial factors be the cause of the high variance? To study this hypothesis we measure the correlation between two runs that share the same network initialization and initial experience, but otherwise have independent randomness (from e.g. the environment, action sampling, and mini-batch sampling). Across all tasks, we repeat this process for ten combinations of initializations and initial experience, from each such combination we start two runs $A, B$ which do not share other sources of randomness. For each time-step we measure the Pearson correlation between $score(A)$ and $score(B)$. This correlation, averaged across all timesteps, is shown in Figure 2 (left). We see that the correlation is very small, suggesting that the initialization and initial experience does not have a large effect on the outcome. Specifically, the hypothesis that there are good and bad initializations that leads to good or bad performance is inconsistent with the results of Figure 2 (left). Correlation across time is noisy but centers around zero, see Appendix A.

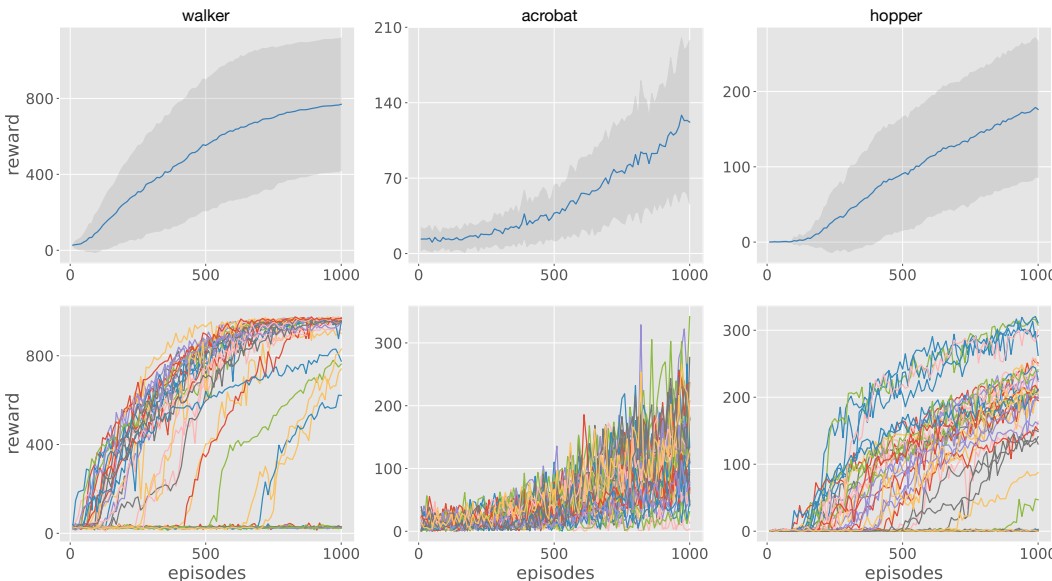

Figure 1: Learning curves over 40 seeds for the baseline agent. The upper row shows the mean reward, with one standard deviation indicated by shading. The bottom row shows individual runs. For some tasks, the variance is dominated by a few outlier runs that completely fail to learn. This is especially clear in the walker task. Removing such outlier runs would decrease the variance.

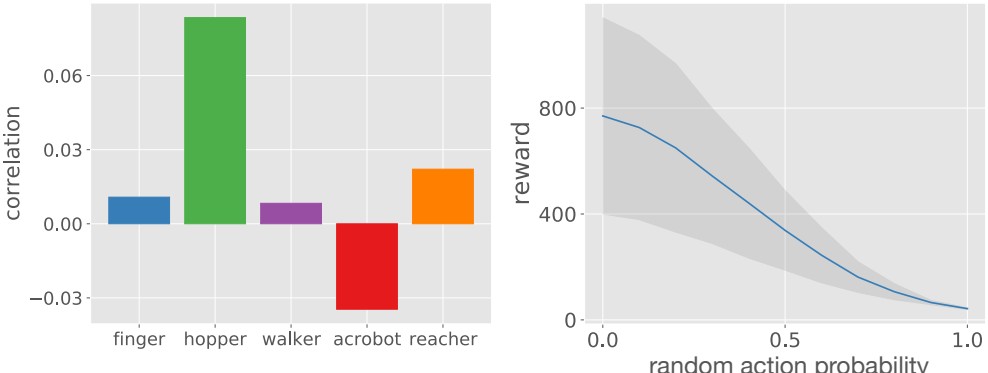

Figure 2: (**Left**) Correlation between runs that share network initialization and starting experience, but otherwise have independent other sources of randomness (e.g. from mini-batch sampling). Across all tasks, the correlation is small. This suggests that initialization and starting experience have limited effect on the outcomes. (**Right**) The average reward for the walker task when interpolating between a learned policy (random action probability = 0) and random policy (random action probability = 1). The rewards obtained decrease gracefully as we adopt a more random policy. This suggests that the environment is relatively forgiving with respect to actions.

### 3.3    ARE THE ENVIRONMENTS TOO SENSITIVE?

Another cause of variance is the environments. Perhaps the environments require very specific policies to yield rewards, and unless the agent stumbles upon a good policy early on, learning fails. To study the sensitivity of the environments we consider gradually exchanging a good learned policy with a random policy. Specifically, at every timestep $t$, with probability $p$, we exchange the policy's action with a random action. We use the final policy of a performant run with the baseline agent as the learned policy. The effect of this, for ten final policies learned for the walker environment, is shown in Figure 2 (right). Performance degrades gracefully as we gradually move to a random policy. This suggests that the environment is relatively stable with regard to actions. Results for other environments are similar and are shown in Appendix A. Furthermore, in Appendix E we verify that the variance arising from evaluating the agent on a finite number of episodes is small compared to the inter-run variance. Thus, environment sensitivity and evaluation methodology do not appear to explain the excessive variance.

### 3.4    IS FEATURE LEARNING TO BLAME?

In the early stages of training, the image encoder will extract poor features. Without good representations, it is hard to learn a good policy, and without a policy generating interesting behavior it is hard to learn good representations. If such a vicious cycle was causing runs to fail, jump-starting the encoder to extract good features should help to decrease the variance. To test this hypothesis, we consider initializing the weights of the convolutional encoder to some good pretrained values before training as normal. For each task, we pick a run with above-average final performance and use its convolutional weights at episode 500. We refer to this strategy as pretrain. We also consider the same strategy but decrease the learning rate of the pre-trained weights by a factor of 10x. We refer to this strategy as pretrain + small lr. In addition to this fix, we also consider improving the features with self-supervised learning during early training (Jing & Tian, 2020). We use the loss function proposed in Grill et al. (2020) which is useful for smaller batch sizes (we use 256) and has already successfully been applied in RL (Schwarzer et al., 2021). We use standard methods in self-supervised learning: a target network with exponentially moving weights and a *head* network. See Appendix A for details. We consider adding the self-supervised loss to the first 10,000 and 20,000 steps and refer to these modifications as self-supervised 10k and self-supervised 20k. The results for this experiment are shown in Table 1. For these fixes, the large variance persists even though the performance improves marginally with self-supervised learning. We thus hypothesize that representation learning is not the main bottleneck. Indeed, experiments in later sections of the paper show that one can significantly decrease variance without improving representation learning.

Table 1: Performance when using self-supervised learning or pretrained weights for the concolutional encoder. Improvements are small and variance remain high. This suggests that poor features might not cause the high variance. Indeed, Table 2 shows that performance can be improved significantly without better representation learning. Metrics are calculated over 10 seeds.

| metric | method | acrobot | finger turn | hopper hop | reacher | walker | avg. |
|---|---|---|---|---|---|---|---|
| $\mu$ | baseline | 122.0 | 278.6 | 176.2 | 678.0 | 769.1 | 404.8 |
| $\mu$ | self-supervised 10k | 141.4 | 456.2 | 176.5 | 671.6 | 677.1 | 424.6 |
| $\mu$ | self-supervised 20k | 159.4 | 325.3 | 209.5 | 659.6 | 773.8 | **425.5** |
| $\mu$ | pretrain | 160.0 | 303.7 | 153.3 | 620.8 | 581.4 | 363.8 |
| $\mu$ | pretrain + small lr | 157.3 | 358.8 | 124.8 | 679.0 | 674.2 | 398.8 |
| $\sigma$ | baseline | 76.0 | 221.2 | 90.3 | 167.3 | 350.1 | 181.0 |
| $\sigma$ | self-supervised 10k | 68.4 | 276.1 | 104.6 | 222.4 | 425.9 | 219.5 |
| $\sigma$ | self-supervised 20k | 67.7 | 191.9 | 71.6 | 164.8 | 374.4 | **174.1** |
| $\sigma$ | pretrain | 39.8 | 146.7 | 86.8 | 168.7 | 429.1 | 174.2 |
| $\sigma$ | pretrain + small lr | 71.8 | 204.6 | 102.7 | 165.4 | 393.5 | 187.6 |

## 3.5 STABILITY OF NETWORK PARAMETRIZATION

We have considered the effects of initialization, feature learning, and environment sensitivity and found that none of these factors satisfactorily explain the observed variance. Another factor to consider is the stability of the network parametrization. Unstable parametrization could make the learning process chaotic and thus cause high variance. To study this hypothesis we investigate the failure modes of individual runs. In Figure 3 we consider some runs that fail as well as some successful runs for the walker task. Plots for all runs are available in Appendix A. We plot three quantities during training: rewards, actor gradient norm $\|\nabla\ell\|$, and average absolute value of actions $|\mathbf{a}|$. All agents initially have poor rewards, but once the rewards start to improve, they improve steadily. During this initial period of no or little learning, the gradients of the actor-network (row two in Figure 3) are small. Recall that the actor is optimized via gradients: $\nabla_\theta Q_\phi(\mathbf{s}, \mathbf{a}_\theta(\mathbf{s})) =$

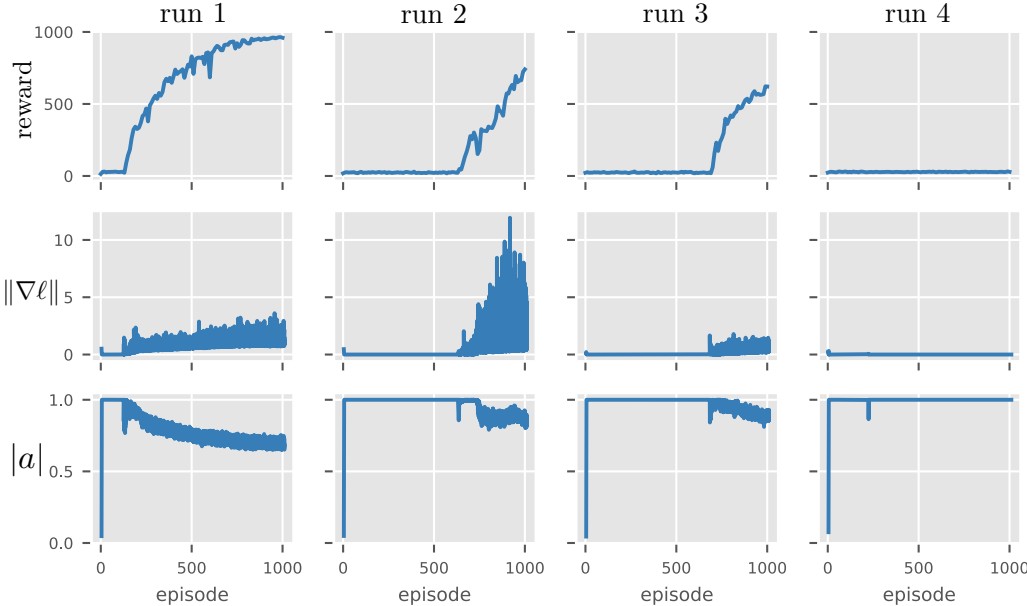

Figure 3: Each row shows a different quantity during training for the baseline agent. Row 1 shows rewards, row 2 shows actor gradient norm $\|\nabla\ell\|$, and row 3 shows average absolute action $|\mathbf{a}|$. Each column corresponds to an independent run of the walker task: run 1 is successful, runs 2 and 3 eventually learn, while run 4 fails to learn. Before learning starts, all runs suffer from small gradients. During this period, the average absolute value of actions is close to 1. This causes small gradients since the gradient of $\tanh(x)$ is small when $|\tanh(x)| \approx 1$.

$\sum_i \frac{\partial Q_\phi}{\partial \mathbf{a}_i} \frac{\partial \mathbf{a}_i}{\partial \theta}$. If this gradient is close to zero, we might suspect that $\frac{\partial \mathbf{a}_i}{\partial \theta} \approx 0$. This could easily happen if the `tanh` non-linearity in equation 2 is saturated. In row three of Figure 3 we show the average absolute value of the actions, which are all bound to lie in $[-1, 1]$. We indeed see that for the runs that fail the average absolute action stays around 1 throughout training, which implies that the gradients for the actor are small since $|\tanh(x)| \approx 1$ implies $\frac{\partial}{\partial x} \tanh(x) \approx 0$. The cause of saturating tanh must be excessively large activations, i.e. $\mathbf{a}_\theta^{pre}(\mathbf{s})$ in equation 2 becoming too large, which saturates the tanh non-linearity. In fact, all runs seem to initially suffer from saturating tanh, but most runs escape from this poor policy. Somewhat similarly, Wang et al. (2020) has observed that saturating nonlinearities can impede exploration when removing the entropy bonus from SAC agents. However, for the DDPG agent saturation does not harm exploration since the exploration noise is added outside the non-linearity as per eq. (2). Instead, the issue here appears to be vanishing gradients which impedes learning. The Q-values could of course also suffer from exploding activations, but since there are no saturating nonlinearities for the Q-values, such issues are harder to pinpoint. These experiments suggest that avoiding excessively large activations could reduce the number of poor outlier runs, and thus decrease variance.

## 4 IMPROVING STABILITY

Motivated by our qualitative observations we now consider a few ideas for improving the stability of learning, hoping that these will decrease the variance. We consider several methods:

**Normalizing activations.** As per eq. (2), the actor selects actions using $\mathbf{a}_\theta^{pre}(\mathbf{s})$, where $\mathbf{a}_\theta^{pre}(\mathbf{s})$ is a deep neural network. Figure 3 shows that learning can fail due to excessively large activations. To avoid this we propose to parametrize $\mathbf{a}_\theta^{pre}(\mathbf{s})$ as a linear function of features with a fixed magnitude. Specifically, let $\Lambda_\theta(\mathbf{s})$ be the penultimate features of the actor-network. We propose to simply normalize $\Lambda_\theta^n(\mathbf{s}) = \Lambda_\theta(\mathbf{s})/\|\Lambda_\theta(\mathbf{s})\|$ and let $\mathbf{a}^{pre}$ be a linear function of $\Lambda_\theta^n$, i.e. $\mathbf{a}_\theta^{pre}(\mathbf{s}) = L\Lambda_\theta^n(\mathbf{s})$ for a linear function $L$. Note that no mean subtraction is performed. We refer to this straightforward modification as *penultimate normalization* or *pnorm* for short. When applied to the actor, we simply call the strategy *actor pnorm*. Our motivation has been to stabilize the actor output, but the critic network could also benefit from the stability offered by normalizing penultimate features. We refer to the strategy of normalizing both the actor and the critic as *both pnorm*. We also consider *layer norm* (Ba et al., 2016) applied similarly to both the actor and critic. We also consider the normalization scheme of Wang et al. (2020), which directly constrains the output of the actor. We refer to this method as *output norm*. At last, following Gogianu et al. (2021), we consider adding *spectral* normalization (Miyato et al., 2018) to the penultimate layer.

**Penalizing saturating actions.** Another strategy for avoiding actions with large magnitude is to add a term to the loss function which directly penalizes large actions. One can imagine many such loss functions, we here consider the simple and natural choice of squared loss $\ell(\theta, \mathbf{s}) = \lambda\|\mathbf{a}_\theta^{pre}(\mathbf{s})\|^2$. We can simply add the square loss times some hyperparameter $\lambda$ to the loss function. By tuning $\lambda$, one should be able to remove the saturating actions illustrated in Figure 3. We heuristically find that $\lambda = 0.000001$ works well in practice (see Appendix A) and use this value. We refer to this modification as *penalty*. Since the Adam optimizer is invariant under re-scaling the loss, the specific value of $\lambda$ has a small effect as long as $\nabla_\theta Q_\phi(\mathbf{s}, \mathbf{a}_\theta(\mathbf{s})) \approx 0$, which happen as per Figure 3.

**Learning rate warmup.** One common strategy to improve stability is to start the learning rate at zero and increase it linearly to its top value during early training (Goyal et al., 2017). This can improve numerical stability (Ma & Yarats, 2021). We consider using this strategy and linearly increasing the learning rate from zero over the first 10,000 steps. We refer to this strategy as *lr warmup*.

**Small weights final layer.** Following Andrychowicz et al. (2020), we consider using smaller weights for the final layers of the actor and the critic. Specifically, we downscale these weights by a factor of 100 at initialization time. We refer to this strategy as *scale down*.

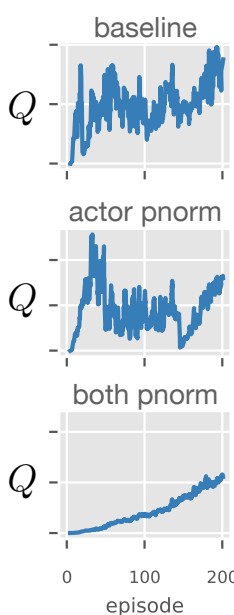

Figure 4: Average Q-value during training.

Table 2: Performance for various stability improvements. These methods often decrease variance, especially for the walker task where variance decreases by orders of magnitude. Larger stability often leads to higher reward, likely as poor outlier runs are eliminated. Normalizing features gives consistent improvements. Metrics are calculated over 10 seeds.

| metric | method | acrobot | finger turn | hopper hop | reacher | walker | avg. |
|--------|--------|---------|-------------|-----------|---------|--------|------|
| $\mu$ | baseline | 122.0 | 278.6 | 176.2 | 678.0 | 769.1 | 404.8 |
| $\mu$ | penalty | 128.0 | 431.6 | 241.7 | 584.6 | 956.3 | 468.4 |
| $\mu$ | actor pnorm | 117.8 | 372.7 | 240.5 | 701.7 | 956.7 | 477.9 |
| $\mu$ | both pnorm | 192.2 | 605.8 | 188.7 | 892.7 | 961.5 | **568.2** |
| $\mu$ | layer norm | 43.5 | 661.2 | 218.6 | 659.7 | 952.4 | 507.1 |
| $\mu$ | lr warmup | 150.3 | 265.4 | 173.7 | 660.9 | 670.4 | 384.1 |
| $\mu$ | grad clip 1 | 138.2 | 214.0 | 206.0 | 520.0 | 675.6 | 350.7 |
| $\mu$ | grad clip 10 | 79.3 | 166.2 | 157.7 | 548.4 | 678.7 | 326.0 |
| $\mu$ | spectral | 104.0 | 101.0 | 201.0 | 213.6 | 775.6 | 279.1 |
| $\mu$ | scale down | 127.2 | 258.4 | 111.7 | 508.7 | 386.4 | 278.5 |
| $\mu$ | output norm | 13.3 | 306.3 | 216.9 | 699.4 | 952.1 | 437.6 |
| $\sigma$ | baseline | 76.0 | 221.2 | 90.3 | 167.3 | 350.1 | 181.0 |
| $\sigma$ | penalty | 42.1 | 160.6 | 46.5 | 236.8 | 15.4 | 100.3 |
| $\sigma$ | actor pnorm | 39.9 | 227.9 | 42.7 | 101.9 | 8.0 | **84.1** |
| $\sigma$ | both pnorm | 60.1 | 193.4 | 78.4 | 122.5 | 7.0 | 92.3 |
| $\sigma$ | layer norm | 57.2 | 228.4 | 112.5 | 297.1 | 13.5 | 141.7 |
| $\sigma$ | lr warmup | 53.8 | 133.6 | 98.2 | 133.4 | 423.4 | 168.5 |
| $\sigma$ | grad clip 1 | 60.6 | 127.1 | 92.4 | 231.9 | 425.3 | 187.5 |
| $\sigma$ | grad clip 10 | 56.1 | 143.9 | 113.6 | 243.2 | 426.3 | 196.6 |
| $\sigma$ | spectral | 37.0 | 61.4 | 75.0 | 224.8 | 128.0 | 105.2 |
| $\sigma$ | scale down | 27.8 | 186.9 | 96.6 | 241.3 | 444.0 | 199.3 |
| $\sigma$ | ourput norm | 16.6 | 190.4 | 21.4 | 180.4 | 17.2 | 85.2 |

**Gradient clipping.** Another stabilizing strategy is gradient clipping (Zhang et al., 2020). To avoid occasional exploding gradients, one simply clips the norm of gradients that are above some threshold. This strategy is popular in NLP (Gehring et al., 2017; Peters et al., 2018) and is sometimes used in RL (Raffin et al., 2019). We consider clipping the gradient when the norm, calculated independently for the actor, critic, and convolutional encoder, is larger than 1 or 10. We refer to these strategies as *grad clip 1* and *grad clip 10*.

**Results.** In Table 2 we evaluate the effects of these ideas. Penultimate normalization has the best performance and lowest variability on average, whereas layer norm can make performance worse. Since normalizing the critic improves performance, the critic might also suffer from instabilities. This instability can be observed by simply plotting the average Q-values during training for various agents, see Figure 4. We study this question in detail in Appendix C. The penalty method improves the performance while many other methods decrease average reward, and lowering variance by using a poor policy is not hard to achieve and not particularly desirable. The output normalization method of Wang et al. (2020) performs worse than actor normalization, and importantly fails completely on the acrobot task. We hypothesize that directly constraining the output space is harmful on the acrobot task. These experiments show that it is possible to significantly decrease the variance of the baseline agent by applying stabilization methods. Additionally, decreasing the variance often increases the average reward – likely as bad outlier runs are eliminated.

## 4.1 COMBINING FIXES

We have found that many of the proposed methods can decrease variance and improve average reward. For optimal performance, we now consider combining three orthogonal methods which improve performance individually – actor and critic penultimate normalization, pre-tanh penalty, and self-supervised learning (we apply it to the first 10,000 steps). In Table 3 we compare the performance between the baseline and an agent using these three modifications (which we refer to as *combined*) over 40 seeds. We see that the variances go down while the average score goes up. In fact, for the walker task, the variance decreases by orders of magnitude. Eliminating poor

Table 3: Performance when combining three improvements: penultimate normalization, action penalty and early self-supervised learning. The variance decreases and the performance improves across all tasks. This demonstrates that one can substantially decrease variance in RL without hurting average reward. Metrics are calculated over 40 seeds.

| metric | method | acrobot | finger turn | hopper hop | reacher | walker | avg. |
|--------|--------|---------|-------------|------------|---------|--------|------|
| $\mu$ | baseline | 122.0 | 278.6 | 176.2 | 678.0 | 769.1 | 404.8 |
| $\mu$ | combined | 227.7 | 653.5 | 231.1 | 893.4 | 964.2 | **594.0** |
| $\sigma$ | baseline | 76.0 | 221.2 | 90.3 | 167.3 | 350.1 | 181.0 |
| $\sigma$ | combined | 59.0 | 183.1 | 40.8 | 120.0 | 6.2 | **81.8** |

outlier runs seems to benefit the average performance in addition to decreasing variance. While our methods have mostly been motivated by outlier runs that fail catastrophically, they likely help more normal runs too. Learning curves for the combined agent are given in Appendix A. In Appendix D, we verify that our fixes enable the agent to learn successfully with a larger learning rate.

## 4.2 ADDITIONAL TASKS

We now apply the combined agent of Table 3 to additional tasks. Specifically, we consider all 21 tasks identified as easy or medium by Yarats et al. (2021b). Note that the combined agent has been constructed using only 5 of these tasks, so this experiment measures generalization. For computational reasons, the additional 16 tasks only use 10 seeds each. We use the baseline hyperparameters across all tasks without tuning (Yarats et al. (2021b) tunes parameters on a per-task basis). Results are shown in Figure 5. Our methods consistently improve the average reward while the average standard deviation decreases by a factor $> 2$. Furthermore, a few additional tasks (finger spin, hopper stand, walker run, walker stand) have their variance decreased by an order of magnitude or more. The tasks cheetah run and quadruped walk perform worse than the baseline, although this could potentially be alleviated by tuning hyperparameters. Just increasing the learning rate on these tasks improve performance, see Figure 13 in Appendix A. These results demonstrate that our modifications significantly decrease variance and that directly optimizing variance can be feasible and fruitful. Performance profiles (Agarwal et al., 2021) are available in Figure 16 of Appendix A.

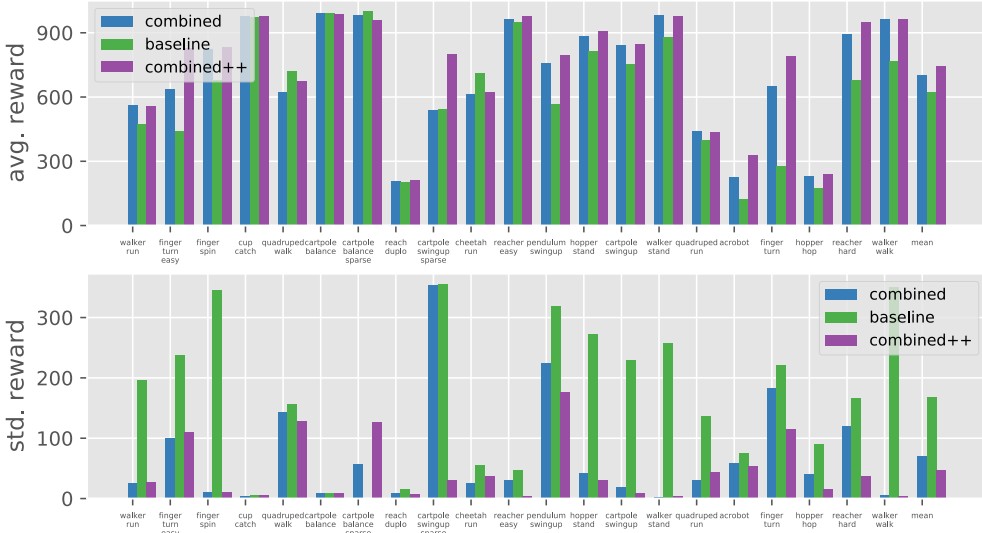

Figure 5: Performance for 21 tasks. (**Top**) Average reward per task. The combined agents improve the average reward, and most tasks benefit from our modifications. (**Bottom**) Standard deviation of reward, broken up by task. The combined agents have less variability, and for a few tasks the variance decreases by orders of magnitude. Except for measurements in Table 3, we use 10 seeds.

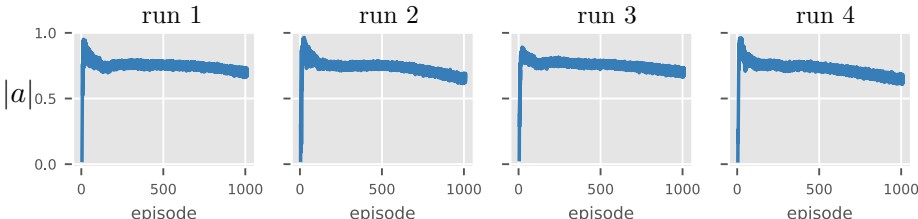

Figure 6: The average absolute value of the actions for the combined agent. We show four representative runs of walker task. Compared to the baseline algorithm in Figure 3, the actions do not saturate early in training despite increasing rapidly, indicating improved stability.

### 4.3  SPARSE TASKS

Consider the five tasks in Figure 5 where the combined agent has the highest standard deviation. Four out of these have sparse rewards – finger turn hard, reacher hard, pendulum swingup, and cartpole swingup sparse. Once stability has been addressed, sparsity is a major driver of residual variance. Motivated by this, we consider three fixes that could help for sparse tasks. Firstly, not training until at least one non-zero reward has been observed. Secondly, extending the self-supervised training throughout the whole training process. Thirdly, exchanging the clipped double Q-learning $Q = \min(Q_1, Q_2)$ (Fujimoto et al., 2018) with $Q = \text{avg}(Q_1, Q_2)$, to decrease pessimistic bias. Through ablations (see Appendix B), we find that doing this for the critic loss (eq. (1)) but retaining the clipping $Q = \min(Q_1, Q_2)$ for the actor loss performs the best. We dub this strategy *asymmetrically clipped double* Q-learning. In Appendix B we present an ablation study over these three modifications, finding that using more self-supervised learning and asymmetrically clipped double Q-learning improves performance. We add these two modifications to our combined agent, which we now call combined++. In Figure 5 we show the performance of this agent (seeds are independent from experiments in Appendix B). Performance improves, especially for sparse tasks, and compared to the baseline agent the average standard deviation decreases by a factor $> 3$.

## 5  EPILOGUE

**Related Work.** There are multiple papers documenting the high variance of RL (Henderson et al., 2018), focusing on implementation details (Engstrom et al., 2019), continuous control (Mania et al., 2018), Atari games (Clary et al., 2018) and real-world robots (Lynnerup et al., 2020). Given this high variance, there is ample work on how to accurately measure performance in RL (Colas et al., 2019; Chan et al., 2020; Jordan et al., 2020; Agarwal et al., 2021; Islam et al., 2017). Our strategy of simply reducing the variance of RL algorithms is complementary to developing robust metrics. Within Q-learning, there is a large amount of work aiming to improve and stabilize Q-estimates (Fujimoto et al., 2018; Nikishin et al., 2018; Anschel et al., 2017; Jia et al., 2020; Chen et al., 2018). These apply to the discrete DQN setting, distinct from continuous control. There is also work on stabilizing RL for specific architectures (Parisotto et al., 2020) and task types (Kumar et al., 2019; Mao et al., 2019). Normalization in RL have been investigated by Salimans & Kingma (2016); Bjorck et al. (2021b); Gogianu et al. (2021), and Wang et al. (2020). Wang et al. (2020) directly normalizes the output of the actor, which e.g. for a one-dimensional task shrinks the action space to $[\tanh(-1), \tanh(1)]$ (modulo noise). As per Table 2, this method underperforms our methods.

**Limitations.** We have only considered a single SOTA agent (DRQv2 of (Yarats et al., 2021b)) on one popular suite of tasks. Furthermore, we focus on an agent with tanh nonlinearities for the policy. While this is an important agent class, other important agent types (Hafner et al., 2021) and environments (Bellemare et al., 2013) remain. We show that *it is possible* to decrease variance with simple modifications, although which methods are effective likely depends upon the base algorithm. While many of our fixes can potentially be added to other agents, it remains to be experimentally tested if doing so improves performance. We leave this question for future work. While our fixes typically are only a few lines of code, they nonetheless increase the agent complexity.

**Conclusion.** We have investigated variance in a popular continuous control setup and proposed several methods to decrease it. We demonstrate across 21 tasks that these dramatically decrease variance while improving average rewards for a state-of-the-art agent. Our experiments show that directly optimizing for lower variance can be both feasible and fruitful.

**Ethics Statement.** There are many positive RL applications – examples include safe autonomous cars, accelerated scientific discovery, and medical testing. Studying causes of high variance, and proposing fixes to this issue, can benefit real-world adoption of RL techniques. Additionally, reducing the variance of RL experiments would aid the research community as reproducibility can be problematic in the field. However, there are also many potentially harmful military applications. Our study is academic in nature and only considers simulated toy environments, but insights from our work could nonetheless be useful in harmful applications. We do not believe that our work differs from other RL work significantly concerning ethics.

**Reproducability Statement.** In Appendix A we detail the infrastructure and software used. The perhaps most important component of generating reproducible findings in RL is to use many seeds. To this end, we have focused on five environments but evaluated these thoroughly. The main experiments of the paper (Table 3) use 40 seeds whereas all others use 10 seeds.

**Acknowledgement.** This research is supported in part by the grants from the National Science Foundation (III-1618134, III-1526012, IIS1149882, IIS-1724282, and TRIPODS- 1740822), the Office of Naval Research DOD (N00014- 17-1-2175), Bill and Melinda Gates Foundation. We are thankful for generous support by Zillow and SAP America Inc. This material is based upon work supported by the National Science Foundation under Grant Number CCF-1522054. We are also grateful for support through grant AFOSR-MURI (FA9550-18-1-0136). Any opinions, findings, conclusions, or recommendations expressed here are those of the authors and do not necessarily reflect the views of the sponsors. We thank Rich Bernstein, Wenting Zhao, and Ziwei Liu for help with the manuscript. At last, we thank our anonymous reviewers for fruitful interactions.

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

# A APPENDIX

**Tuning** $\lambda$. For tuning the hyperparameters $\lambda$ we search over the set $\{1e-1, 1e-2, 1e-3, 1e-4, 1e-5, 1e-6\}$. For each value, we run the hopper task for a short amount of time with five different seeds. We found that too large $\lambda$ penalized actions too much, and that even very small values like $1e-6$ sufficed to give improvement.

**Details on self-supervised Learning.** We extract features from the convolutional network and the first feedforward layer of the critic, which projects the feature map to 50-dimensions. Feature vectors are normalized to unit norm before being fed into the self-supervised loss (Bachman et al., 2019). We use exponentially moving weights for the target network (He et al., 2020), using the same decay parameters as for the target critic. We use a *head* for the online network which is simply a two-layer MLP with hidden dimension 512 and final dimension 50 (Chen et al., 2020). The two augmentations we consider are simply two adjacent frames with augmentations that the DDPG agent of Yarats et al. (2021b) considers naturally. We use the square loss as proposed in Grill et al. (2020), which has been shown to be useful in smaller batch sizes. The self-supervised loss is then simply added to the normal loss function for the critic.

**Infrastructure.** We run our experiments on Nvidia Tesla V100 GPUs and Intel Xeon CPUs. The GPUs use CUDA 11.1 and CUDNN 8.0.0.5. We use PyTorch 1.9.0 and python 3.8.10. Our experiments are based upon the open-source DRQv2 implementation of Yarats et al. (2021b).

Table 4: Variance across tasks as reported by Yarats et al. (2021b). Mean reward and task variance often increase in tandem – to avoid bias towards easy tasks with high mean and thus high variance, we rank tasks based upon relative variance $\sigma/\mu$. We use the five tasks with the highest relative variance, removing one duplicate: finger turn easy/hard. Used tasks are highlighted. Scores are measured just before 1 million frames (i.e. frame 980000) since not all runs have scores at the 1 million mark. All numbers are rounded to two decimal digits.

| task | $\sigma/\mu$ | $\mu$ | $\sigma$ |
|---|---|---|---|
| finger turn hard | 0.74 | 268.90 | 198.36 |
| walker walk | 0.48 | 769.66 | 371.48 |
| hopper hop | 0.36 | 221.13 | 80.63 |
| finger turn easy | 0.31 | 558.52 | 174.19 |
| acrobot swingup | 0.31 | 177.65 | 54.77 |
| reacher hard | 0.22 | 628.29 | 135.57 |
| walker run | 0.20 | 538.59 | 110.02 |
| cup catch | 0.13 | 909.95 | 118.06 |
| finger spin | 0.12 | 860.30 | 104.18 |
| quadruped run | 0.12 | 446.49 | 53.51 |
| quadruped walk | 0.12 | 732.52 | 87.19 |
| cartpole balance sparse | 0.12 | 962.30 | 113.10 |
| reach duplo | 0.08 | 202.51 | 16.99 |
| cartpole swingup sparse | 0.08 | 760.11 | 60.11 |
| cheetah run | 0.07 | 710.24 | 52.43 |
| reacher easy | 0.05 | 931.54 | 50.86 |
| pendulum swingup | 0.04 | 838.49 | 36.38 |
| hopper stand | 0.02 | 917.85 | 20.45 |
| cartpole swingup | 0.02 | 864.68 | 15.47 |
| walker stand | 0.01 | 980.72 | 5.21 |
| cartpole balance | 0.01 | 993.45 | 5.00 |

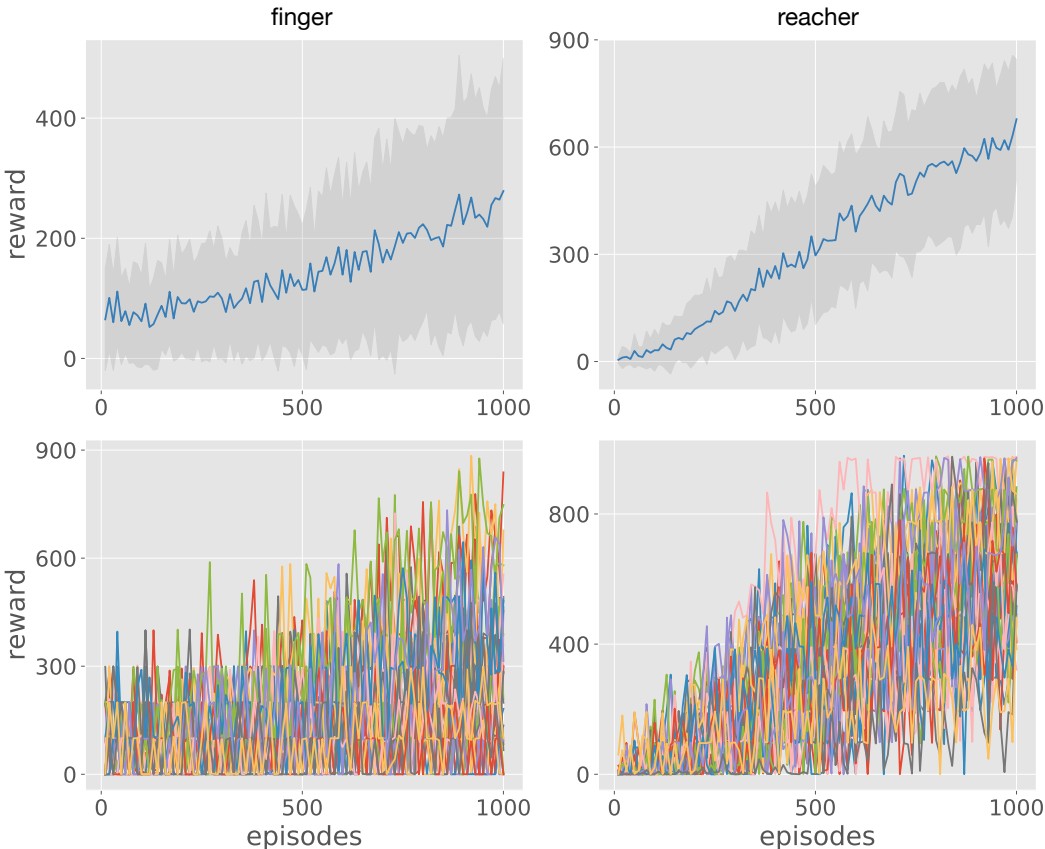

Figure 7: Learning curves for the two tasks not shown in Figure 1. The upper row shows the mean reward with one standard deviation indicated by shading. The bottom row show learning curves for individual runs. Although the signal is noisy, several outlier runs fail to learn.

Table 5: Hyperparameters used throughout the paper. These follow Yarats et al. (2021b) for the *medium* tasks.

| parameter | value |
|---|---|
| learning rate | 1e−4 |
| soft-update $\tau$ | 1e−2 |
| update frequency | 2 |
| stddev. schedule | linear(1.0, 0.1, 500000) |
| stddev. clip | 0.3 |
| action repeat | 2 |
| seed frames | 4e3 |
| batch size | 256 |
| $n$-step returns | 3 |
| discount $\gamma$ | 0.99 |

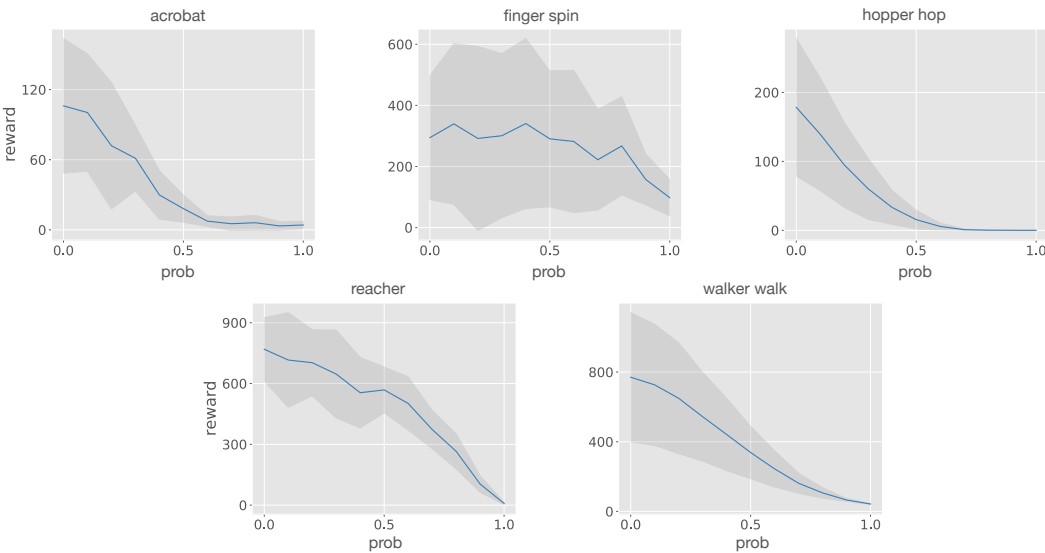

Figure 8: Rewards obtained when gradually replacing a learned policy with a random policy. At every time-step $t$, the action suggested by the learned policy is replaced with a random action with probability $p$. We start from a final policy learned by the baseline algorithm and average over ten such policies. Across all tasks, the rewards decrease gracefully with a more random policy, suggesting that the environments are relatively robust to perturbed policies.

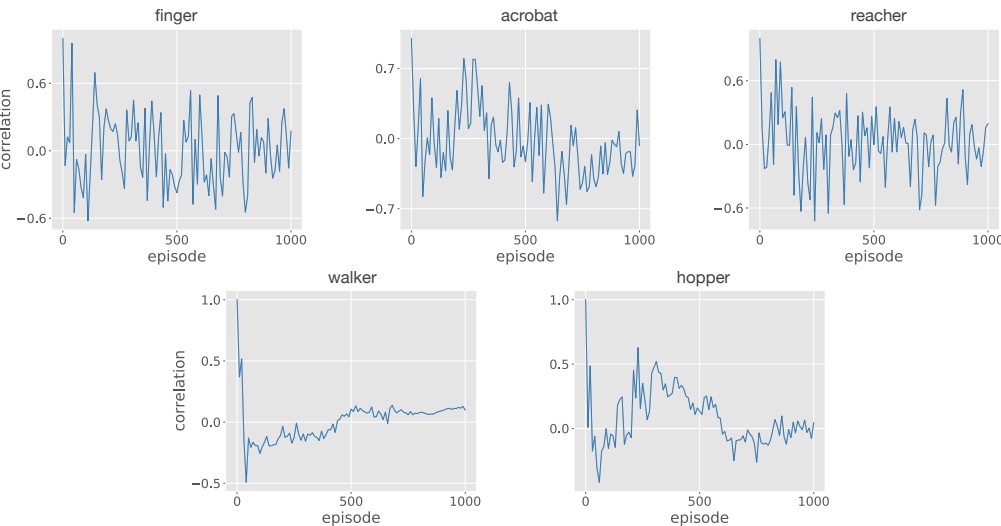

Figure 9: Correlation between the evaluation scores of two runs that share network initialization and seed experience, but otherwise have different randomness. Correlation is calculated across ten runs and plotted against time. The correlation is noisy for many environments but oscillates around zero, suggesting that there is little actual correlation and that network initialization and seed experience have a small effect on performance.

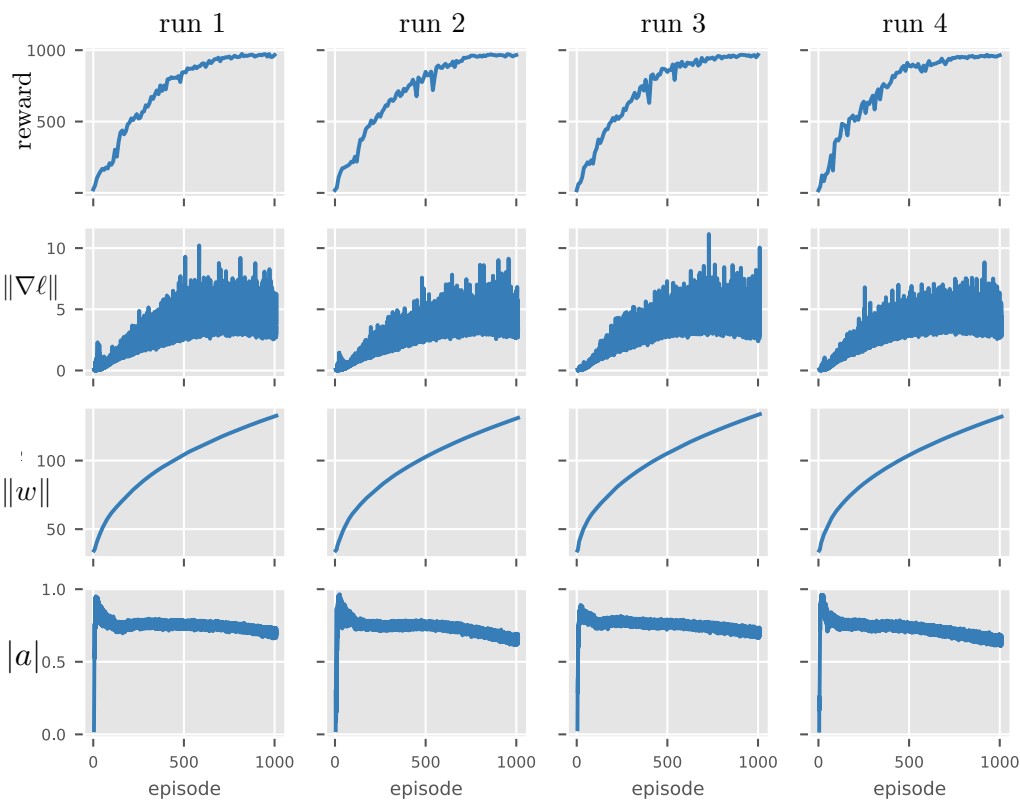

Figure 10: Four quantities shown during training for the combined agent. We show reward, norm of actor gradient $\|\nabla\ell\|$, norm of actor weight $\|w\|$ and average absolute action $|\mathbf{a}|$. The actions never saturate and learning starts quickly. Compare with Figure 3, which shows the same quantities for the baseline algorithm.

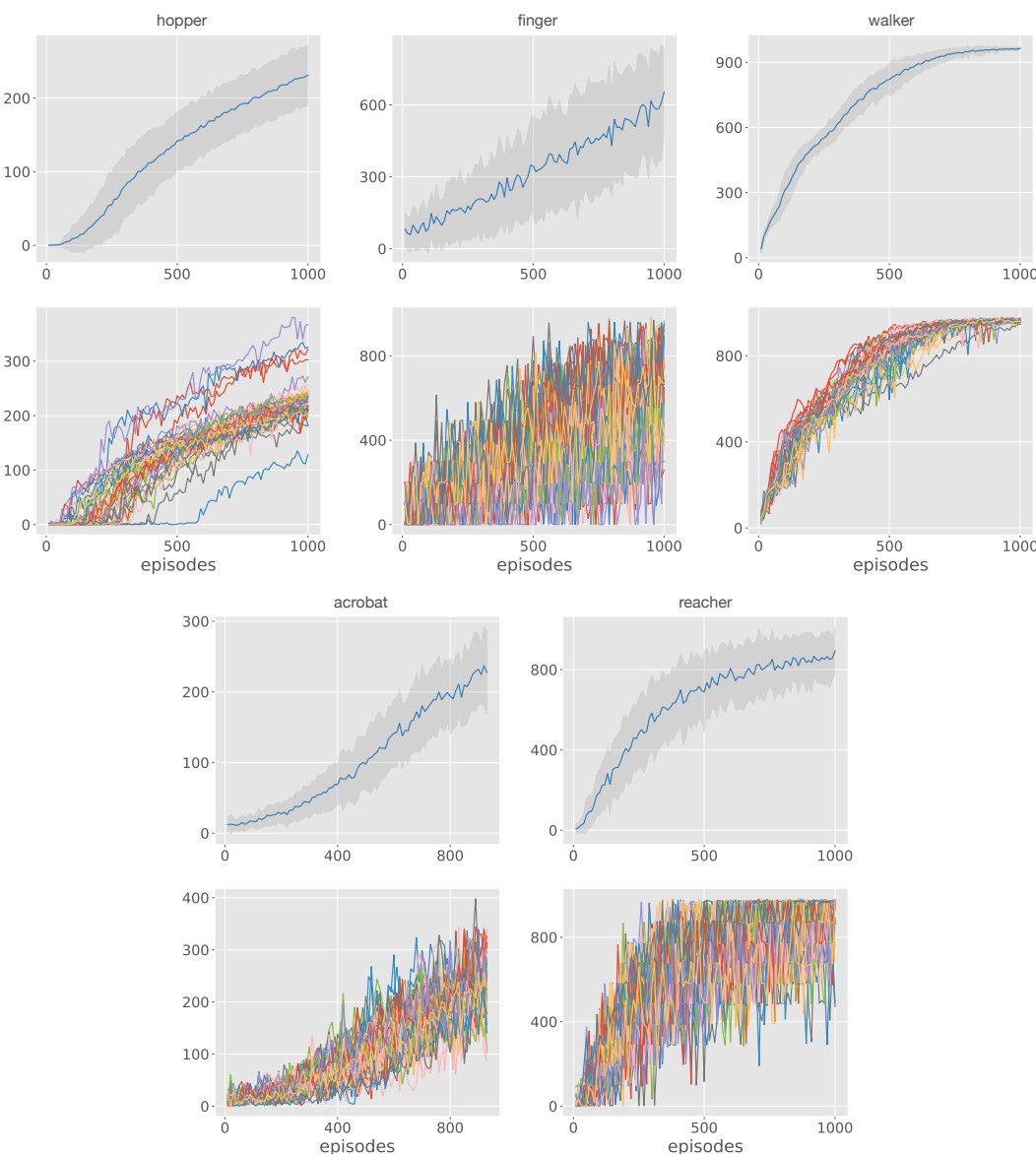

Figure 11: Learning curves when using the combined agent. We show mean reward with one standard deviation indicated by shading and learning curves for individual runs.

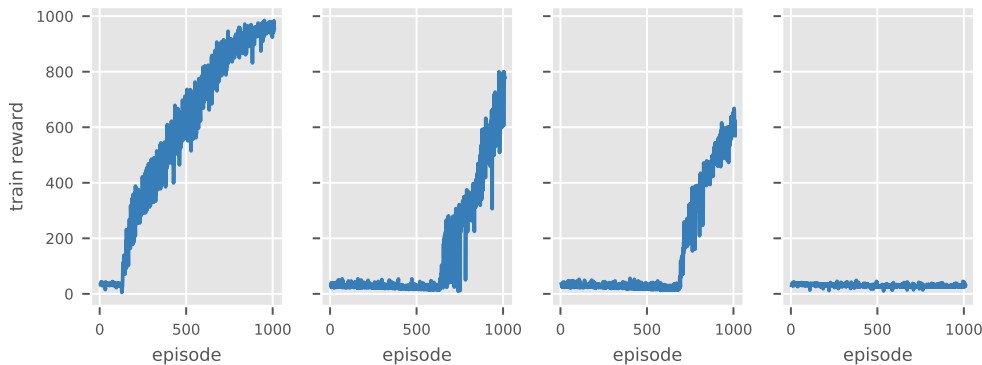

Figure 12: Rewards obtained during training for the runs illustrated in Figure 3. Note that the rewards are nonzero from the start of training. This implies that the agent observes rewards and can train on these. Thus, lack of rewards is not the cause of poor performance.

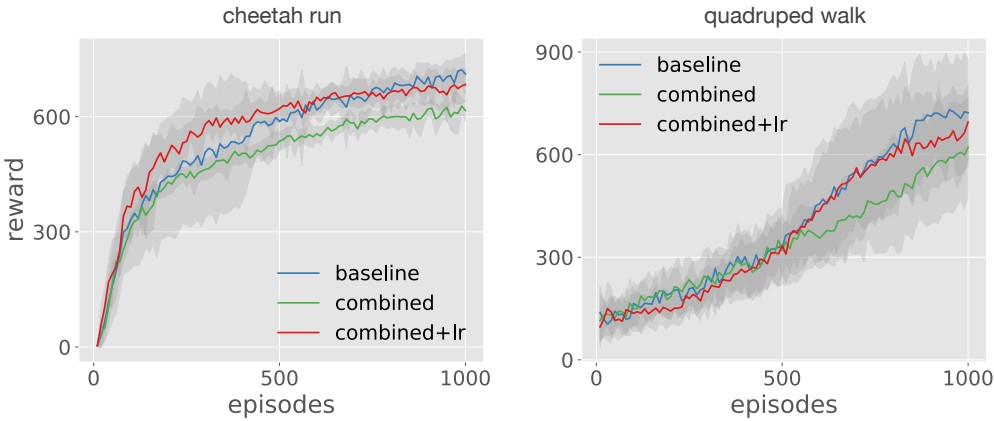

Figure 13: The performance of the combined agents on two tasks when increasing the learning rate by a factor of 10. Performance improves markedly on these tasks. This highlights how hyperparameter tuning could further improve the results of Figure 5.

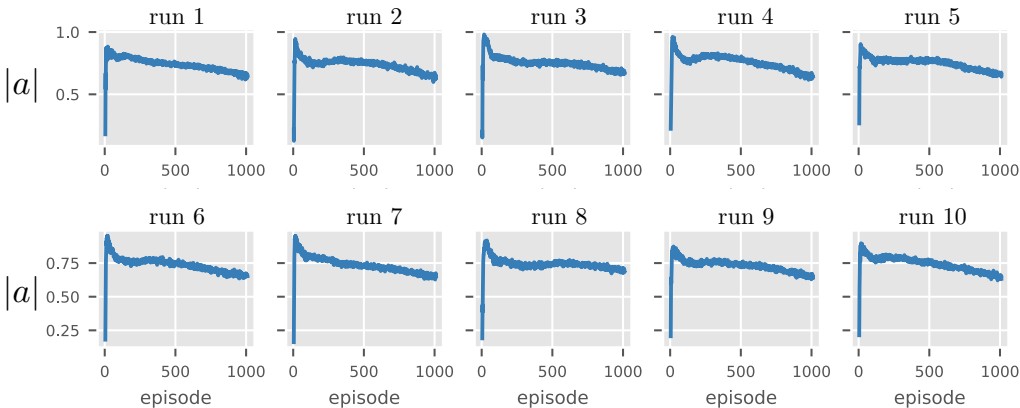

Figure 14: We repeat Figure 6 for all 10 seeds of the both pnorm agent of Table 2. Note that the action distribution does not saturate.

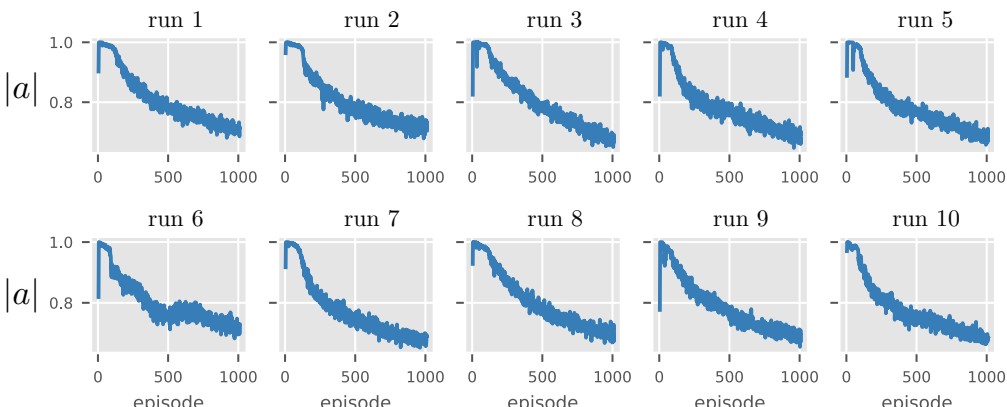

Figure 15: We repeat Figure 6 for all 10 seeds of the penalty agent of Table 2. Note that the action distribution does not saturate.

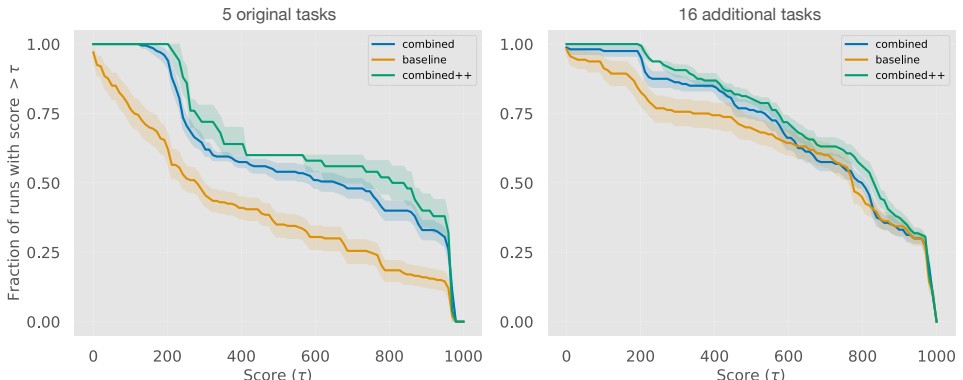

Figure 16: Performance profiles as computer by the rliable library of Agarwal et al. (2021). (**Left**) profiles over the original 5 tasks of Table 3, with 10 seeds for combined++ and 40 seeds for the two other methods. (**Right**) profiles over the additional 16 tasks of Figure 5, with 10 seeds per method. Our combined agents improve performance, especially for the lowest quantiles. The effect is most dramatic for the 5 original tasks, this is natural since we initially selected these as they suffered from high variance. Also note that the baseline is the recent SOTA DRQv2 agent of Yarats et al. (2021b), which is very competitive – so any improvements to it can be very significant.

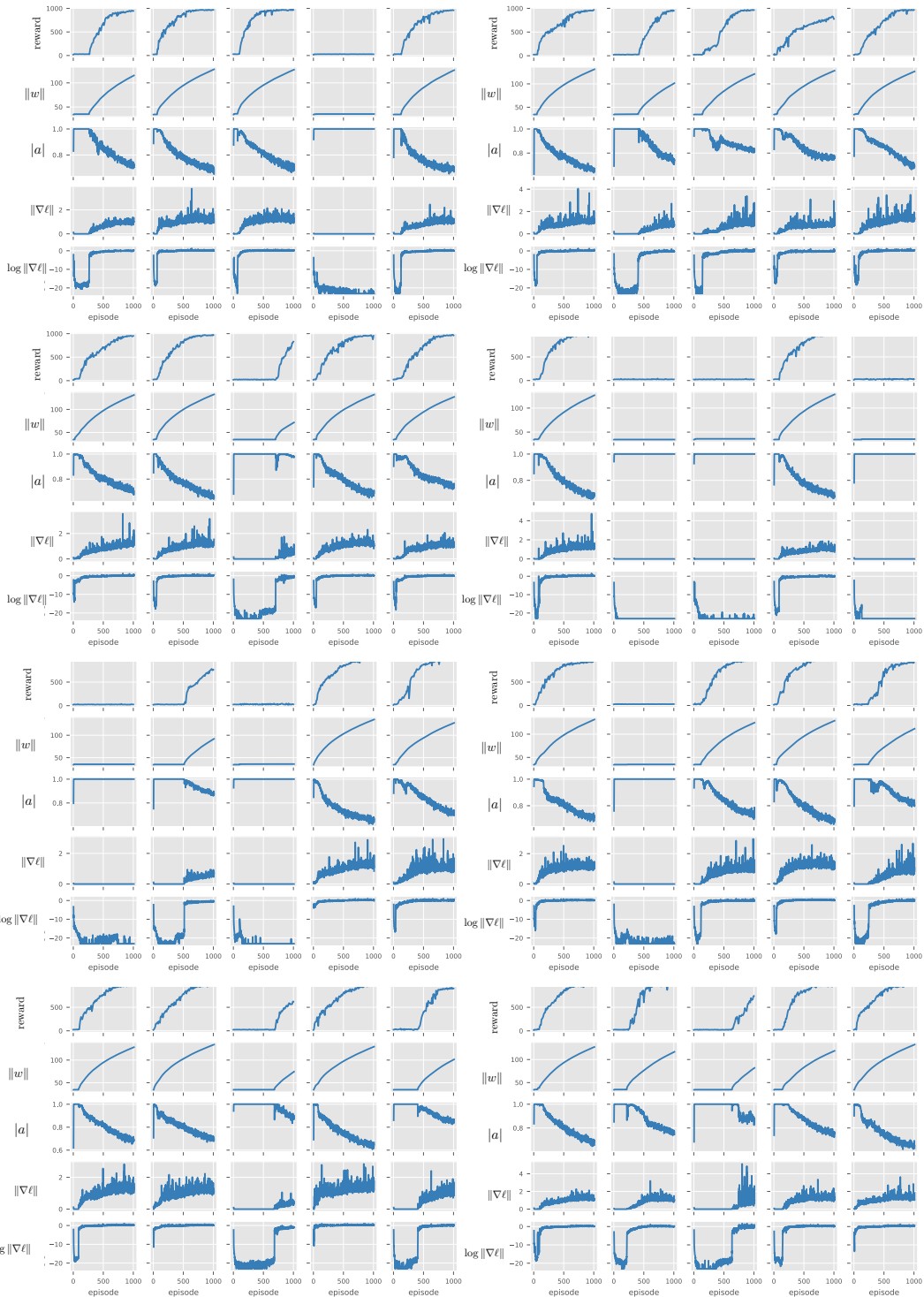

Figure 17: We repeat Figure 3 for all 40 seeds. The plots show five quantities for the baseline agent of the walker task: reward, absolute average action $|a|$, actor weight norm $\|w\|$, actor gradient norm $\|\nabla\ell\|$ and the logarithm of the actor gradient norm $\log\|\nabla\ell\|$. We add $1\mathrm{e}-10$ to the logarithm for numerical stability. Results are similar to Figure 3.

## B IMPROVEMENTS FOR SPARSE TASKS

In Figure 5, consider the five tasks where the combined agent suffers from the highest standard deviation. Four of these tasks have sparse rewards. This suggests that sparsity is a major driver of variance. To further decrease the variance, we now investigate fixes for four sparse tasks – cartpole swingup sparse, pendulum swingup, reacher hard, and finger turn hard. We will consider three additional modifications which aim to specifically combat reward sparsity:

**Avoiding training without rewards.** For the sparse tasks, an agent might not see any rewards during e.g. the first 100 episodes. Perhaps training in the absence of any rewards harms the network weights. The fix we consider is simple, do not use any gradient updates before at least one non-zero reward has been observed.

**Increasing self-supervised learning.** Since the agent might potentially not observe any rewards for 100 episodes, we consider extending the self-supervised learning to ensure some supervisory signal. We simply extend the self-supervised time from 10k steps to the entire training process.

**Removing clipped double q-learning.** The agent of Yarats et al. (2021a) employs clipped double q-learning (Fujimoto et al., 2018) – i.e. the agent uses two critic networks $Q_1, Q_2$ and takes $Q = \min(Q_1, Q_2)$ to avoid overestimation. We hypothesise that the min-operator causes excessive pessimism in sparse environments where reward often are zero. For example, if $Q_1 = 0$ and $Q_2 > 0$ we have $Q = \min(Q_1, Q_2) = 0$. Thus, if one Q-network outputs only zeros and the rewards are almost only zeros, the critic target of eq. (1) will be mostly zeros and the other Q-network will be encouraged to output mostly zeros too. Thus, the Q-network could potentially get stuck with mostly outputting zeros, an issue shown in Figure 18. We thus propose to use $Q_a = \text{avg}(Q_1, Q_2)$. Through ablation experiments in Table 6, we find that using $Q_a = \text{avg}(Q_1, Q_2)$ only for the critic loss (i.e. eq. (1)) but retaining $Q = \min(Q_1, Q_2)$ for the actor update, works the best. We call this strategy *asymmetrically clipped double* Q-learning.

Results for these experiments are shown in Table 7. We see that decreasing pessimism and increasing self-supervised learning help the most. Increasing self-supervised learning helps the most for cartpole swingup sparse and pendulum swingup. Note that these two tasks were not used in the ablation experiments of Table 2. Additionally, whereas feature learning was not the bottleneck for the baseline agent, it might be the bottleneck for the combined agent. Avoiding training without rewards does not seem to improve performance significantly. We conclude that increased self-supervised learning and asymmetrically clipped double Q-learning improve performance for tasks with sparse rewards. We add these two modifications to the combined agent and call the resulting agent combined++. Its performance across all 21 tasks is shown in Figure 5. It achieves even better results, and its average standard deviation is smaller than the baseline by a factor $> 3$.

Table 6: Performance for various types of clipped double q-learning (Fujimoto et al., 2018). We compare exchanging $Q = \min(Q_1, Q_2)$ with $\text{avg}(Q_1, Q_2)$ at two locations – when computing the critic loss (as in eq. (1)) and when computing the actor loss. The baseline always uses $\min$, avg both uses avg at both locations, and avg critic/actor uses the avg only when defining the critic/actor loss. Using $Q = \text{avg}(Q_1, Q_2)$ only when defining the critic loss performs the best. We call the resulting strategy *asymmetrically clipped double* Q-learning.

| metric | method | cartpole sparse | finger turn hard | pendulum | reacher hard | avg |
|--------|--------|-----------------|------------------|----------|--------------|-----|
| $\mu$ | baseline | 641.8 | 596.5 | 648.0 | 872.6 | 689.7 |
| $\mu$ | avg actor | 80.9 | 422.2 | 776.0 | 672.6 | 487.9 |
| $\mu$ | avg both | 393.0 | 432.4 | 647.6 | 548.4 | 505.4 |
| $\mu$ | avg critic | 761.5 | 811.3 | 639.0 | 939.2 | **787.8** |
| $\sigma$ | baseline | 321.5 | 292.4 | 283.5 | 95.5 | 248.2 |
| $\sigma$ | avg actor | 242.6 | 258.8 | 138.9 | 164.8 | 201.3 |
| $\sigma$ | avg both | 393.3 | 171.9 | 323.1 | 312.0 | 300.1 |
| $\sigma$ | avg critic | 127.5 | 89.4 | 318.6 | 37.3 | **143.2** |

Table 7: Performance for three fixes for sparse tasks: asymmetrically clipped double Q-learning (ac), no learning until at least one non-zero reward has been observed (nz) and more self-supervised learning (ssl). We show mean reward and standard deviation of rewards, using ten seeds. Using asymmetrically clipped double Q-learning and more self-supervised learning are the best fixes. Increasing the self-supervised learning helps the most for cartpole swingup sparse and pendulum swingup, two tasks which were not part of the five tasks we consider first.

| metric | method | cartpole sparse | finger turn | pendulum | reacher hard |
|--------|--------|-----------------|-------------|----------|--------------|
| $\mu$ | baseline | 641.8 | 596.5 | 648.0 | 872.6 |
| $\mu$ | ssl+ac | **802.5** | 788.0 | **828.0** | 930.2 |
| $\mu$ | ssl+nz | 607.6 | 659.4 | 813.5 | 869.3 |
| $\mu$ | ac+nz | 561.4 | 770.6 | 624.6 | 860.9 |
| $\mu$ | nz | 618.2 | 663.4 | 657.2 | 882.9 |
| $\mu$ | ac | 761.5 | **811.3** | 639.0 | **939.2** |
| $\mu$ | ssl | 235.0 | 574.6 | 686.7 | 882.7 |
| $\mu$ | ssl+nz +ac | 742.7 | 729.8 | 758.0 | 922.7 |
| $\sigma$ | baseline | 321.5 | 292.4 | 283.5 | 95.5 |
| $\sigma$ | ssl+ac | **29.4** | 135.6 | **24.2** | 47.6 |
| $\sigma$ | ssl+nz | 286.1 | 173.4 | 29.8 | 78.4 |
| $\sigma$ | ac+nz | 306.2 | 150.4 | 330.0 | 80.3 |
| $\sigma$ | nz | 267.5 | 192.1 | 252.5 | 117.7 |
| $\sigma$ | ac | 127.5 | **89.4** | 318.6 | **37.3** |
| $\sigma$ | ssl | 359.7 | 236.7 | 272.5 | 80.1 |
| $\sigma$ | ssl+nz +ac | 151.2 | 130.9 | 220.8 | 45.3 |

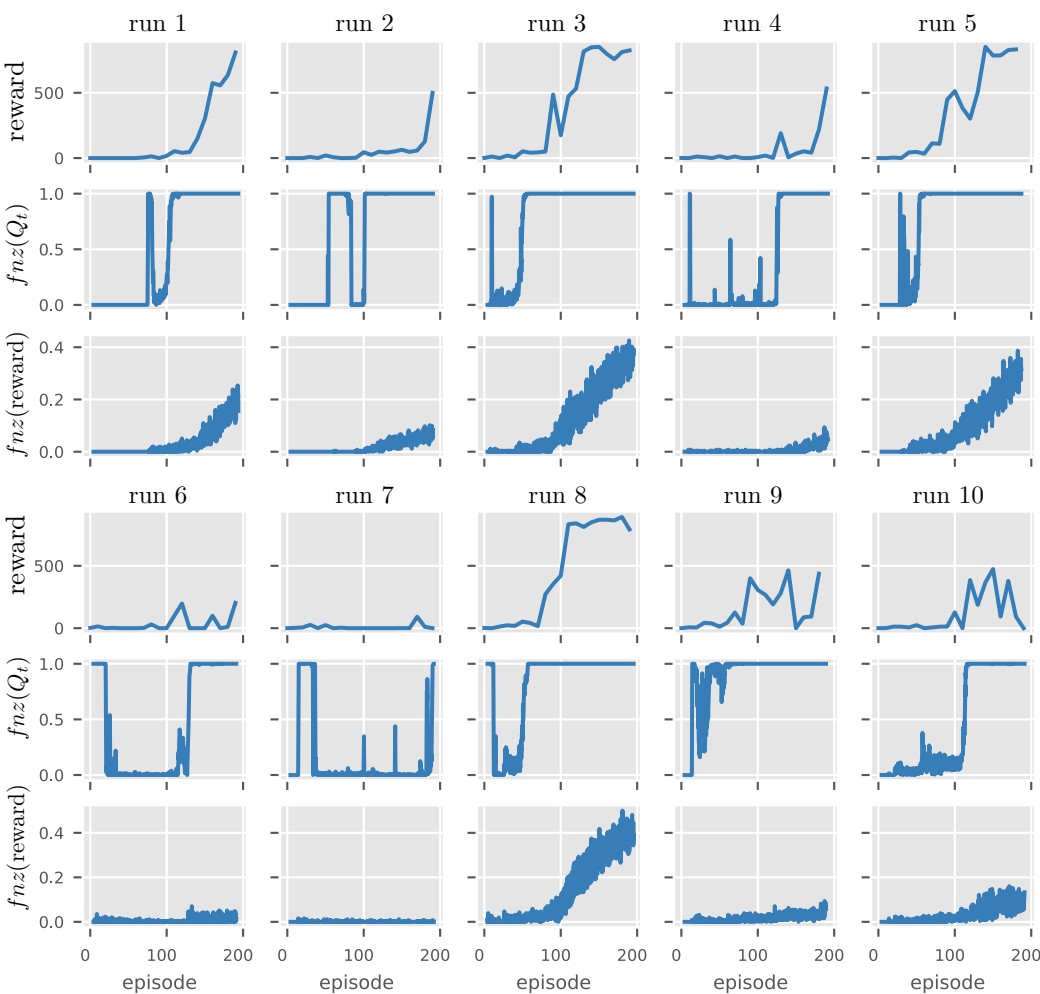

Figure 18: We illustrate three quantities for 10 runs of the pendulum swingup task using the combined agent: the reward, $fnz(Q_t)$ the fraction of non-zero Q-target values obtained over the training batches, and $fnz(\text{reward})$ the fraction of non-zero rewards obtained over the training batches. As shown in the two middle rows, the Q-values frequently "get stuck" at outputting almost only zeros.

## C    NORMALIZATION FOR THE CRITIC

We now study why penultimate normalization is helpful for the critic network. We consider three agents from Table 2: baseline, actor pnorm, and both pnorm. The both pnorm agent uses penultimate normalization for the critic, whereas the others do not. We plot two additional quantities during early training for the reacher hard task, using 10 new seeds. In Figure 19 we show the average Q-values, measured over training batches, achieved by actions generated by the actor network. We see that the curves are much smoother when normalizing the critic, whereas they oscillate aggressively otherwise. Quickly oscillating Q-values makes it hard for the actor to learn a good policy. We hypothesize that stabilizing the Q-values over time is the primary mechanism by which critic normalization helps. In Figure 20 we show the average absolute difference between Q-values before and after a gradient update, as measured over training batches. We see that normalizing the critic often leads to Q-value changes which are smaller and more stable over time. This in turn likely stabilizes learning as per Figure 19. Whereas the Q-value changes can trivially be made smaller by decreasing the learning rate, changes in Q-values depend non-linearly on the learning rate. Furthermore, too low learning rates often hurt generalization (Li et al., 2019). For increased legibility, the curves are smoothed by averaging with a window of width ten.

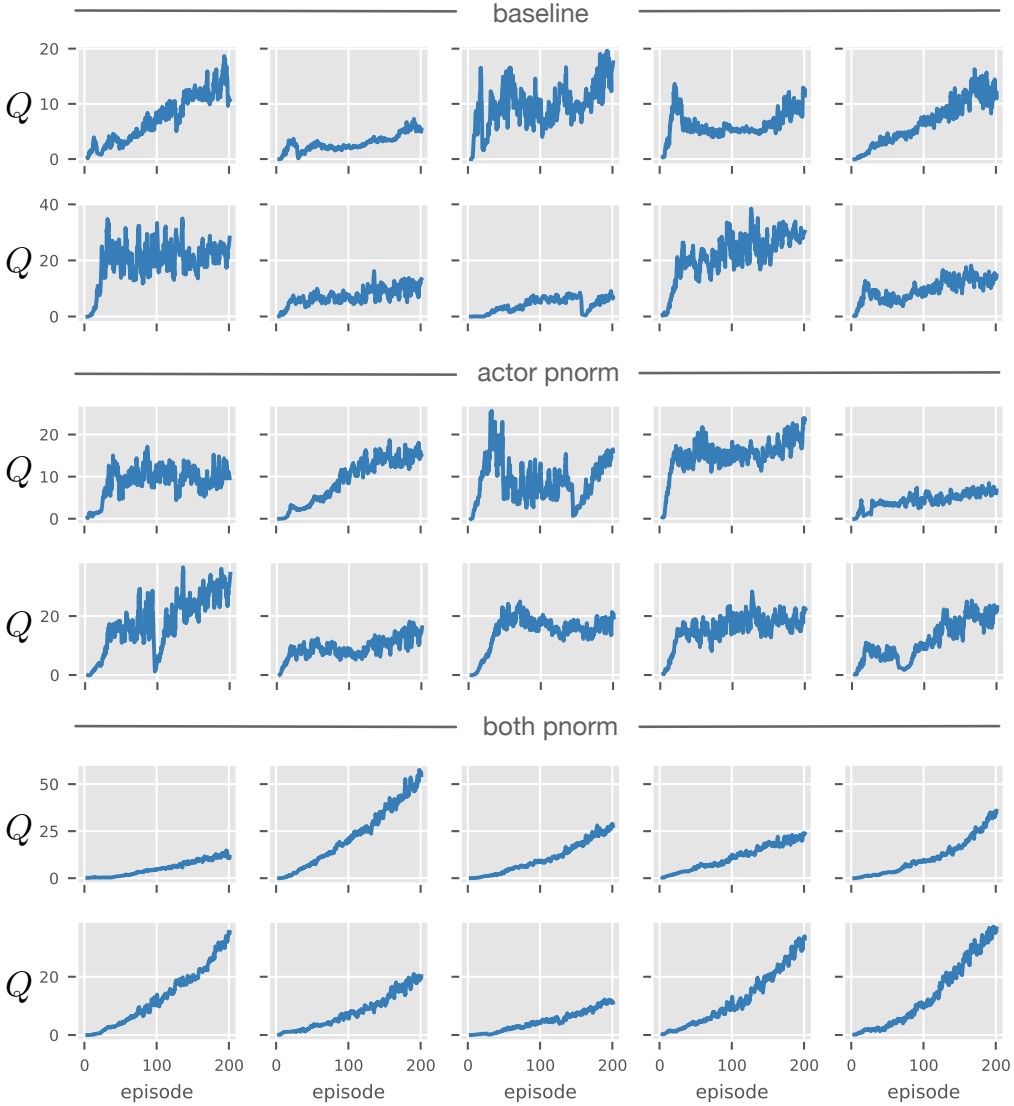

Figure 19: Average Q-values during early training for three agents on the reacher hard task. The both pnorm agent, which uses penultimate normalization for the critic, has Q-values that grow steadily during training. The other agents have Q-values which often fluctuate wildly during training. This suggests that penultimate normalization for the critic stabilizes Q-values, which provides a less noisy signal to the actor.

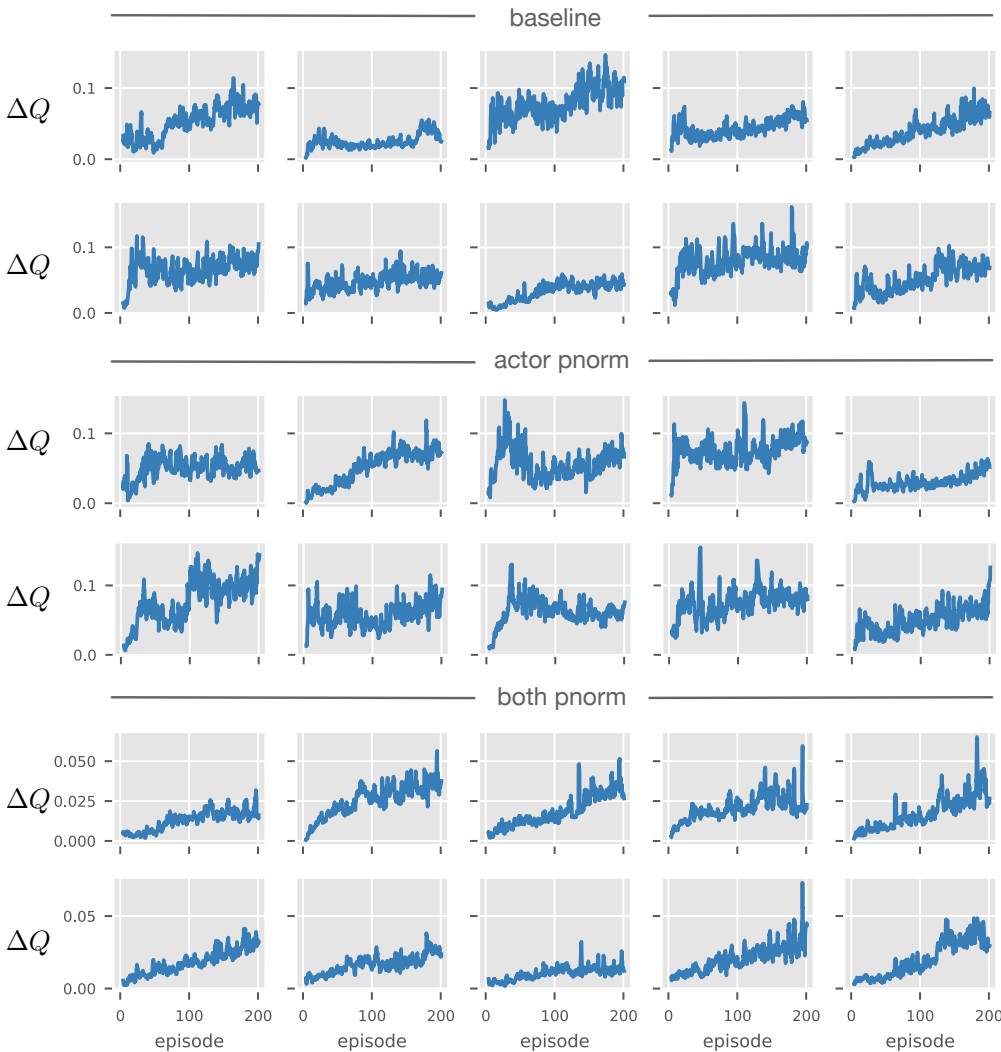

Figure 20: Average absolute difference $\Delta Q$ between Q-values before and after a gradient step. The Q-value changes are generally smaller and more stable over time for the both pnorm agent (which uses penultimate normalization for the critic). This likely stabilizes the Q-values as per Figure 19.

# D    LEARNING RATE STABILITY

To verify that our fixes increase the stability we now consider changing the learning rate. A high learning rate can cause divergence (Li et al., 2019), and learning rate stability can thus serve as a proxy for learning stability. We consider learning rates in the set $\{1e-4, 3e-4, 1e-3, 3e-3\}$, where $1e-4$ is the default learning rate used by Yarats et al. (2021b). Results for the walker task are shown in Table 8. We see that the baseline algorithm fails for large learning rates, the average rewards drop quickly while variance decreases when no effective policy is learned. We show the same experiment for additional tasks, but only one learning rate, in Table 9. Again, we see that the agent can tolerate much higher learning rates when using our modifications.

Table 8: Performance when changing learning rates. The baseline algorithm fails with larger learning rates whereas the combined agent succeeds. This suggests that our fixes improve the stability. Metrics are calculated over 10 seeds on the walker task.

| metric | method | $1e-4$ | $3e-4$ | $1e-3$ | $3e-3$ |
|---|---|---|---|---|---|
| $\mu$ | baseline | 769.1 | 330.2 | 26.8 | 25.5 |
| $\mu$ | combined | 964.2 | 962.2 | 954.9 | 960.1 |
| $\sigma$ | baseline | 350.1 | 415.3 | 3.9 | 6.4 |
| $\sigma$ | combined | 6.2 | 4.7 | 5.6 | 5.1 |

Table 9: Performance for the combined agent and the baseline when using learning rate of $1e-3$. The combined agent works well even with high learning rates, whereas the baseline fails. Statistics are computed over 10 seeds.

| metric | method | acrobot | finger turn | hopper hop | reacher | avg |
|---|---|---|---|---|---|---|
| $\mu$ | baseline | 27.8 | 50.1 | 0.0 | 1.4 | 19.8 |
| $\mu$ | combined | 137.4 | 618.6 | 227.0 | 727.2 | 427.6 |
| $\sigma$ | baseline | 35.6 | 49.9 | 0.0 | 2.0 | 21.9 |
| $\sigma$ | combined | 33.9 | 176.6 | 24.1 | 151.0 | 96.4 |

# E   EVALUATION VARIANCE

We now consider the variance arising from evaluating RL agents on a finite set of episodes. Similar to the bias-variance decomposition, we will see that the variance here can be divided into one part that depends upon the randomness of the algorithm, and one term that depends on the randomness from the finite set of episodes we evaluate on. We let $R$ be a set of training runs and $S$ be a set of evaluation episodes sampled. The performance is measured by

$$\text{perf} = \frac{1}{|R||S|} \sum_{i \in R} \sum_{j \in S} r_{ij}$$

Let us calculate the variance of this quantity. We will assume that rewards from different runs are independent, and that rewards from the same training run are independent conditioned on that run. We now have

$$\text{Var}\left( \frac{1}{|R||S|} \sum_{i \in R} \sum_{j \in S} r_{ij} \right)$$

The variance of a sum of independent variables is simply the sum of their variances. Thus we have

$$\text{Var}\left( \frac{1}{|R||S|} \sum_{i \in R} \sum_{j \in S} r_{ij} \right) = \frac{1}{|R|} \text{Var}\left( \frac{1}{|S|} \sum_{j \in S} r_{ij} \right)$$

Let $\mu = \mathbb{E}[r_{ij}]$. We then have

$$\text{Var}\left( \frac{1}{|S|} \sum_{j \in S} r_{ij} \right) = \mathbb{E}_{alg,samples}\left[ \left( \frac{1}{|S|} \sum_{j \in S} r_{ij} - \mu \right)^2 \right]$$

Here we explicitly calculate the expectation over random training runs (*alg*) and the random episodes sampled (*samples*). We can condition the expectation upon the random training run and then integrate over this expectation. Furthermore, let $\mu_i = \mathbb{E}[r_{ij}|i]$, i.e. the expected value an episode sampled from training run $i$. We then have

$$\text{Var}\left( \frac{1}{|S|} \sum_{j \in S} r_{ij} \right) = \mathbb{E}_{alg}\left[ \mathbb{E}_{samples}\left[ \left( \frac{1}{|S|} \sum_{j \in S} r_{ij} - \mu \right)^2 \Big| i \right] \right]$$

$$= \mathbb{E}_{alg}\left[ \mathbb{E}_{samples}\left[ \left( \frac{1}{|S|} \sum_{j \in S} r_{ij} - \mu_i + \mu_i - \mu \right)^2 \Big| i \right] \right]$$

$$= \mathbb{E}_{alg}\left[ \mathbb{E}_{samples}\left[ \left( \frac{1}{|S|} \sum_{j \in S} r_{ij} - \mu_i \right)^2 \Big| i \right] \right.$$

$$+ 2\mathbb{E}_{samples}\left[ \left( \frac{1}{|S|} \sum_{j \in S} r_{ij} - \mu_i \right)\left( \mu_i - \mu \right) \Big| i \right]$$

$$\left. + \mathbb{E}_{samples}\left[ \left( \mu_i - \mu \right)^2 \Big| i \right] \right]$$

The middle part vanishes since $\mu_i = \mathbb{E}[r_{ij}|i]$. For the first term, given a fixed training run the terms $r_{ij}$ are independent. This gives us

Table 10: Sample variance and algorithm variance as defined in eq. (3). Some tasks have high variance and others have low. However, sampling more episodes does not decrease variance considerable in the 10 training seeds and 10 episodes setting, see eq. (4).

| method | walker | hopper hop | finger turn | acrobot | reacher |
|---|---|---|---|---|---|
| sample | 1922.1 | 450.6 | 143763.0 | 11182.9 | 176211.9 |
| alg | 80002.6 | 9774.0 | 55215.6 | 5710.9 | 35231.7 |

$$\mathbb{E}_{alg}\left[\mathbb{E}_{samples}\left[\left(\frac{1}{|S|}\sum_{j\in S}r_{ij}-\mu_i\right)^2\bigg|i\right]\right]=\frac{1}{|S|^2}\mathbb{E}_{alg}\left[\mathbb{E}_{samples}\left[\left(\sum_{j\in S}r_{ij}-|S|\mu_i\right)^2\bigg|i\right]\right]$$

$$=\frac{1}{|S|^2}\mathbb{E}_{alg}\left[\mathbb{E}_{samples}\left[\sum_{j\in S}\left(r_{ij}-\mu_i\right)^2\bigg|i\right]\right]=\frac{1}{|S|}\mathbb{E}_{alg}\left[\mathbb{E}_{samples}\left[\left(r_{ij}-\mu_i\right)^2\bigg|i\right]\right]$$

We then have

$$\mathrm{Var}(\mathrm{perf})=\frac{1}{|R|}\underbrace{\mathbb{E}_{alg}[(\mu_{alg}-\mu)^2]}_{\text{algorithm variance}}+\frac{1}{|R||S|}\underbrace{\mathbb{E}_{alg}\left[\mathbb{E}_{samples}[(r-\mu_{alg})^2|i]\right]}_{\text{sample variance}} \tag{3}$$

Here $\mu_{alg}=\mu_i$. The first term is what we are interested in measuring. The second term can be large if the environment is noisy and we sample too few episodes. The second term is straightforward to estimate: run the algorithm N times and measure the performance over M samples. For each run, calculate the variance over the samples. Then average this variance over all runs.

In Table 10 we estimate these quantities over ten training runs and ten episodes for the baseline algorithm. As we can see, the environments have very different characteristics. Some have high sample variance and some have very low sample variance. It is common to evaluate with ten training seeds and ten episodes. Can we improve the variance by evaluating on more episodes, say 100? Even for the task with the highest sample noise (reacher), the relative improvement in the standard deviation of the sample variance, i.e.,

$$\sqrt{\left(\sigma_{alg}^2+\frac{1}{10}\sigma_{sample}^2\right)\big/\left(\sigma_{alg}^2+\frac{1}{100}\sigma_{sample}^2\right)}\approx 1.195 \tag{4}$$

is no larger than 20 %. Thus, we conclude that only minor gains can come from evaluating on more episodes. Other environments outside the five we consider, however, might benefit more from using more samples.

## F    EXTENDED RELATED WORK

There are plenty of important applications of RL. Examples include RL for medical testing (Bastani et al., 2021), autonomous vehicles (Bellemare et al., 2020), chemistry (Zhou et al., 2019; Liu et al., 2019), robotics (Kober et al., 2013) and communications infrastructure (Liu et al., 2020). Many more applications remain. Many of these applications require a low variance, but low-power inference (Bjorck et al., 2021a) can also be important for e.g. autonomous robots. Below we discuss some papers closely related to ours in detail.

Engstrom et al. (2019) studies how implementation level details affects performance from policy gradient methods. They specifically find that code-level optimizations can have a significant impact on performance, often rivaling that of algorithmic innovations. This insight agrees with our experimental observations that small modifications can significantly influence performance.

Henderson et al. (2018) presents among the first studies of variability and reproducibility in deep reinforcement learning. They demonstrate that many minor details can have an outsized effect on the outcomes – seeds, environment randomness, and even hardware non-determinism. In our work, we have demonstrated how improved implementations can significantly decrease variance – and our improved agent suffers from much smaller variability between seeds.

Islam et al. (2017) studies variance and reproducibility for continuous control with policy gradient methods. They discuss what metrics are suitable for comparing algorithms and highlight how hyperparameter tuning significantly affects outcomes and thus fairness of evaluations. Our results show that variance in continuous control can be significantly decreased with minor fixes, but we do not study variance from hyperparameter tuning which is an important direction.

Mania et al. (2018) demonstrates that random search is in fact a competitive method for finding policies for continuous control tasks. This strategy is computationally cheap, and the authors present a large study of over 100 seeds, demonstrating that the variance is high. RL still suffers from high variance, but it is not clear if random search still is competitive compared to RL agents.

Clary et al. (2018) studies variability for the Atari benchmark and finds large variability. In light of this, it is proposed that post-training performance should be reported as a distribution rather than as a point estimate. Our paper does not study the Atari suite, but this is an important benchmark that can model discrete tasks rather than continuous robot control problems.

Lynnerup et al. (2020) presents a study over variance for evaluating RL agents on real-world robots. It is concluded that documenting hyperparameters and performing rigorous statistical analysis of the results is imperative. We only consider simulated tasks which are easy to reproduce as long as source code is available, reproducibility becomes much harder in the real world.

Colas et al. (2019) discusses hypotheses testing in the context of RL experiments. They make recommendations regarding significance testing and discuss violations of statistical assumptions. Statistical testing has been criticized in later research (Agarwal et al., 2021), and we refrain from using it.

Chan et al. (2020) proposes multiple metrics to measure notations of reliability in reinforcement learning, and presents an open-sources library to help with this. They discuss the distinction between variability during or after training and the use of significance testing. We have refrained from using hypothesis testing as it suffers from some issues (Agarwal et al., 2021).

Jordan et al. (2020) shows that hyperparameters significantly affect the outcome of RL experiments, and proposes an evaluation strategy that takes this into account. To limit the computational footprint, we have used fixed hyperparameters. While we do not study it, hyperparameter stability is very important for practical applications, and developing RL agents with such stability is an outstanding challenge.

Agarwal et al. (2021) studies variance and statistical significance for RL, focusing especially on Atari tasks where they provide experiments over 100 seeds which shows outlier runs. Among other things, they propose using inter-quartile-means and performance profiles for measuring performance. We adopt the use of performance profiles, but since we are especially interested in outliers inter-quartile-mean is not suitable for our purposes.

Fujimoto et al. (2018) discusses how variance can be harmful to Q-learning, and propose several fixes to it, culminating in the TD3 agent. Some of these fixes have become very popular, especially double clipped Q-values. In Appendix B we revisit double clipped Q-values and find that it can hurt performance for sparse tasks.

Anschel et al. (2017) shows that averaging Q-values can improve the stability and improve performance for a DQN agent. This strategy increases the computational footprint, and such ensembling is known to often work well for supervised learning. The paper mainly focuses on Atari tasks, in contrast, we consider continuous control tasks which arguably are closer to real-world applications.

Nikishin et al. (2018), in a similar spirit, demonstrates that averaging weights can improve the stability of learning. The authors demonstrate that this can improve performance on Atari tasks, although only a handful of seeds appears to be used. To the best of our knowledge, this method is not widely adopted in RL, and we do not experimentally study it.

Jia et al. (2020) considers using classical variance reduction methods such as SVRG (Johnson & Zhang, 2013) to improve deep Q-learning. The authors introduce a recursive framework for computing the stochastic gradient estimate and demonstrate that it can improve the performance of DQN. Again, this method is not widely adopted in RL to the best of our knowledge, and we do not empirically study it.

Chen et al. (2018) considers RL as applied to a dynamic environment, motivated by applications to online recommendation. They propose stratified sampling replay, which relies on a prior customer distribution to decrease variance and the use of approximate regret. These ideas improve performance for DQN agents for a tip recommendation system, while this is an important setting, it is far from continuous control.

Greensmith et al. (2004) studies the use of variance reduction methods for policy gradient methods. Specifically, they study the use of baseline methods, which can significantly decrease the variance and often is adopted in deep reinforcement learning. The use of baselines for policy gradient methods is distinct from the setup we consider.

Parisotto et al. (2020) considers the use of transformers for reinforcement learning and finds that naively using them leads to poor performance. To address this, architectural modifications are proposed that lead to more stable training and improve performance. Studying transformer architectures is important for RL, but in this study, we only consider classical CNN and MLP networks.

Kumar et al. (2019) considers stability for off-policy reinforcement learning, focusing on a setting where the off-policy data is fixed and there is no further interaction with the environment. In this setting, it is demonstrated that bootstrapping error is a key driver of instability, a method to address this issue is proposed. Off-policy reinforcement learning is very important for practical applications, but not the focus of our work.

Mao et al. (2019) considers RL where a stochastic exogenous process influences the environment. In such a setting, a gradient policy method with a standard state-dependent baseline suffers from high variance, and the authors consider fixes to this. The setting considered in this work is distinct from the continuous control tasks we use.

Gogianu et al. (2021) studies the use of spectral normalization (Miyato et al., 2018) for improving the performance of DQN for Atari tasks. They find that this method can stabilize learning when applied to the penultimate layer. Due to the similarity with the normalization methods we consider, we consider this strategy in Table 2.

Wang et al. (2020) studies the behavior of a SAC agent when the entropy bonus is removed. They find that the tanh activations saturate which effectively eliminates exploration since SAC adds exploration noise inside the tanh non-linearity. This argument does not apply to DDPG which adds the exploration noise outside the non-linearity. We study the proposed method Table 2 and find that it underperforms, especially for the acrobot task.

Salimans & Kingma (2016) proposes weight normalization, which reparametrizes the weights by decoupling magnitude and direction. They demonstrate that this idea improves performance in DQN and other settings. Despite the paper being well-known, this strategy has not become widely adopted in RL to the best of our knowledge. Similarly, to the best of our knowledge, batch normalization (Ioffe & Szegedy, 2015; Bjorck et al., 2018) is typically preferred in supervised learning.

