# OpenReview forum: "Is High Variance Unavoidable in RL? A Case Study in Continuous Control"
_ICLR.cc/2022/Conference — ICLR 2022 Poster_

### Official Review · Reviewer_J8Mk · 2021-10-28

**Correctness:** 4
**Technical Novelty And Significance:** 2
**Empirical Novelty And Significance:** 4
**Recommendation:** 6
**Confidence:** 3

**Main Review:**

I believe the problem this paper concerns is very important to the area. High instability in the early training process is a serious issue that stops RL from being more widely applied. I like the authors eliminating possible factors that might cause high variance before finally identifying the main cause. The improvement of the combined agent seems solid. However, I do have the following concerns.

1. It is not clear whether the similar variance-reducing techniques could be applied to other environments. For example, penalizing saturating actions only applies when the actions are continuous variables, which is not the case in Atari games or other environments with discrete action space.

2. I don't fully understand the logic of testing environment sensitivity. The variance is guaranteed to reduce since the mean reward reduces. I think one should compare the variance of the original algorithm to the algorithm with a pre-trained good policy that may automatically eliminate the possibilities of reaching bad states, which may reduce the variance.

3. Some techniques that are used to reduce variance are not new and are commonly used in previous literature like penalizing actions and learning rate warmup.

-------------------------------- Update after rebuttal -----------------------------------------------
I have read the insightful discussion on the newly-discovered work and I appreciate that the authors' responded quickly with new results and pointed out the novelty compared to the new paper. I support accepting the paper.

**Summary Of The Paper:**

This paper gives a systematic evaluation on different factors that may cause the high-variance of actor-critic algorithm on various robotic control problems. They found that the main source of variance is from the numerical instability. The authors then propose four training techniques to mitigate the instability, which significantly reduced the variance.

**Summary Of The Review:**

I work mostly on the theoretical side of RL so I am not sure how significant the experimental results are in this paper.

---

> ### Author Response · Authors · 2021-11-18
> **Review 4**
>
> Thanks for your encouraging response, we are very happy you thought our area is important! We address individual points below:
>
> 1. *“It is not clear whether the similar variance-reducing techniques could be applied to other environments. For example, penalizing saturating actions only applies when the actions are continuous variables, which is not the case in Atari games or other environments with discrete action space.”*
>
> This is a reasonable point, we respond to it in general comment #1.
>
> 2. *“I don't fully understand the logic of testing environment sensitivity. The variance is guaranteed to reduce since the mean reward reduces. I think one should compare the variance of the original algorithm to the algorithm with a pre-trained good policy that may automatically eliminate the possibilities of reaching bad states, which may reduce the variance.”*
>
> We do consider a fixed and good pre-trained policy, and then add noise to it. We have clarified this setup in the text. We don’t fully understand what experiment the reviewer is requesting, would you be able to elaborate? Our point in Figure 2 is that the mean reward is high, even when e.g. 20% of the actions are random while the rest 80% come from a good policy.
>
> 3. *“Some techniques that are used to reduce variance are not new and are commonly used in previous literature like penalizing actions and learning rate warmup.”*
>
> We agree that our methods are not particularly innovative in and of themselves. Our contribution is in identifying a compact set of improvements that robustly decrease the variance of a state-of-the-art method in an important and well-studied setup. Furthermore, this demonstrates that it is fruitful to optimize directly for lower variance, rather than only average reward as is commonly done in empirical RL.
>
> Please let us know if there are any new concerns or additional questions we can respond to!

---

> > ### Comment · Reviewer_J8Mk · 2021-11-25
> > **Update after rebuttal**
> >
> > Thanks for the response. It helped me understand the paper better. I decide to keep my rate but I will increase my Confidence level.

---

### Official Review · Reviewer_ti7w · 2021-11-01

**Correctness:** 3
**Technical Novelty And Significance:** 3
**Empirical Novelty And Significance:** 2
**Recommendation:** 6
**Confidence:** 3

**Main Review:**

Strengths:
 -  While existence of outlier runs is well known in deep RL, for example, see evaluation of 100 seeds on continuous control tasks in [1] (Figure 3)  and 100 seeds on on Atari 100k in [2] (Appendix), the empirical study on continuous control tasks to find what causes such runs is interesting. For example, it seems to rule out network initialization and feature learning.
- The paper suggests a number of simple fixes for control tasks, such as normalizing feature activations in actor and critic, which are easy to implement and often lead to smaller variance on the 5 tasks in the paper.
- The paper is well-written and easy to read.

Weaknesses:

- One of the main limitations of empirical study is that its conclusions are specifically tied to network architectures that use a tanh squashing function in the policy network. For example, policy gradient methods (such as PPO/TRPO) or DQN-style methods typically use `relu` activations and are unlikely to suffer from saturating nonlinearities. The same limitation also applies to the proposed fixes which try to prevent tanh from saturation. This limits the possible impact of this work.

- The visualizations on walker walk seems somewhat misleading as this is the only task where the proposed fixes results in ~60x lower variance. This doesn’t seem to be true for other tasks such as finger turn where the improvements are significantly smaller. Same is true for the experiments for learning stability (can the authors possibly repeat it on other tasks too?) It is unclear if this is the case of showing evidence only on the tasks which support the hypothesis.

- The fix which works well is combining feature normalization for both actor and critic but the rationale for normalizing features in the critic network is not explored in the paper (I believe that the critic network uses ReLU activations only). This makes me wonder if we actually understand why the proposed fix improves performance or reason behind poor outlier runs on tasks other than walker walk?

- The use of a single benchmark with only a small subset of tasks with the high-variance issue casts doubt about the generality of the findings and proposed fixes. I feel lack of compute is not a solid argument to evaluate on only a small number of environments. To give an analogous example from deep RL, [3] show that methods proposed for tackling hard-exploration methods hurt performance on easy-exploration tasks. So, do the proposed fixes actually result in any performance improvement on the remaining tasks used by Yarats et. al (2021b) in the medium tasks. Furthermore, do the combined fixes in Section 4.2 generalize to harder tasks?

- **(Missing result)** The paper introduction mentions reducing variance on a humanoid locomotion task by two orders of magnitude, however I couldn’t find this experiment anywhere.

Other comments (Minor):

- There is a distinction between self-supervised methods like BYOL  and contrastive methods like SimCLR. The authors interchange them throughout the paper which is technically incorrect. Furthermore,  CURL actually hurt performance compared to the baseline method on Atari and their reported improvement resulted from mismatch in evaluation protocols [2] and seems to be a poor example to cite.
- Properly warm-starting / reusing features probably require lowering the learning rate for feature layers so that learning may not lose this initialization quickly after a few updates. Thus, it is unclear if feature learning experiments actually capture this or not.
- Reporting mean and std is a specific kind of interval estimate, so the statement in related work about being different than [2] didn’t compile for me.
- The results in Table 2 were somewhat overwhelming to read because of the plethora of numbers  across multiple rows and columns, maybe reporting aggregate statistics such as mean/median/ interquartile mean with CIs might be better visualization.
- More detailed discussion about outlier runs in RL and discussing this in related work.
- Spelling: intractable → intractable
- While I appreciate the use of a large number of seeds, there is a trade-off between generalizable findings and statistically robust findings. Using a single task with 200 seeds is an extreme example of this.  As such, my opinion is that the conclusions in this work would be more generalizable if the authors used a large number of tasks but maybe slightly smaller number of seeds (maybe 30?) as that provides us with more diverse data points to validate our findings.

[1] Mania, Horia, Aurelia Guy, and Benjamin Recht. "Simple random search of static linear policies is competitive for reinforcement learning."  NeurIPS (2018).

[2] Agarwal, Rishabh, Max Schwarzer, Pablo Samuel Castro, Aaron Courville, and Marc G. Bellemare. "Deep reinforcement learning at the edge of the statistical precipice."  NeurIPS (2021).

[3] Taiga, A.A., Fedus, W., Machado, M.C., Courville, A. and Bellemare, M.G., 2021. On Bonus-Based Exploration Methods in the Arcade Learning Environment. ICLR (2020).

**Summary Of The Paper:**

This paper investigates sources of poor outlier runs in a DDPG-based agent on 5 image-based DM control tasks. The empirical analysis suggests that such outlier runs result from saturating tanh nonlinearities due to large-valued activations in the policy network. To prevent saturation of tanh, the paper proposes and evaluates fixes and shows that combining a bunch of them results in better performance with lower variance.

**Summary Of The Review:**

Although I liked the empirical study, the analysis and the proposed findings may not generalize beyond the setup used by the paper (tanh nonlinearities) and lack of thorough evaluation on a large set of environments. Furthermore, there are other weakness such as missing results and fixes without much motivation with possibility of visualizing results on a cherry-picked environment. Overall, I feel the paper can be significantly improved by addressing some of these weaknesses.


-------------------------------- Update after rebuttal -----------------------------------------------

Based on the discussion with the authors, I feel that the paper can be accepted at the conference. I'd like to update my score from a 5 (weak reject) --> 7 (accept) but the ICLR review scale only allows me to pick a score of 6 or 8. Since a very related work (highlighted by reviewer cHfn is not discussed), I decided to go with a score of 6.

---

> ### Author Response · Authors · 2021-11-18
> **Review 3**
>
> Thanks for your detailed feedback and suggestions for additional experiments!
>
> 1. *“The use of a single benchmark with only a small subset of tasks with the high-variance issue casts doubt about the generality of the findings and proposed fixes. “*
>
> We have added results for all 21 tasks in the {easy, medium} set of [1]. See Figure 4. Over these 21 tasks, our combined agent improves average reward and decreases the average standard deviation by more than half. These large-scale experiments also show that after stabilization has been addressed, reward sparsity drives variance. We further discuss fixes to this.
>
> [1][Yarats et. al. Mastering Visual Continuous Control: Improved Data-Augmented Reinforcement Learning]
>
> 2. *“walker walk [...] is the only task where the proposed fixes results in ~60x lower variance.”*
>
> We have added results for 21 tasks in Figure 4. The variance improves by more than an order of magnitude for multiple tasks. Thus, several tasks have dramatic improvements. As mentioned in the original text, we use the walker task as a running example. Its failure modes are very clear, so this task is easy to interpret.
>
> 3. *“can the authors possibly repeat [the learning rate stability experiments] on other tasks too?”*
>
> We show the performance for two additional tasks when using a higher learning rate in Figure 12. We will add further learning rate stability experiments to our “to-do” list!
>
> 4. *“The paper introduction mentions reducing variance on a humanoid locomotion task by two orders of magnitude, however I couldn’t find this experiment anywhere”*
>
> We refer to the “walker walk” task as a “humanoid locomotion” task. Since the dm-control suite has another task named “humanoid” this understandably caused confusion. The main text now refers to it as a “walking” task.
>
> 5. *“Reporting mean and std is a specific kind of interval estimate, so the statement in related work about being different than [2] didn’t compile for me.”*
>
> We agree. We now only state that “Agarwal et. al. propose new robust estimates of performance”.
>
> 6. *“The results in Table 2 were somewhat overwhelming to read because of the plethora of numbers across multiple rows and columns, maybe reporting aggregate statistics such as mean/median/ interquartile mean with CIs might be better visualization.”*
>
> Thanks for this suggestion! We have added an average score to the table which serves as an aggregate score. Robust metrics such as interquartile mean are unsuitable since we specifically are interested in outliers.
>
> 7. *“More detailed discussion about outlier runs in RL and discussing this in related work.”*
>
> We have expanded the discussion of this in the related work section. We have also added citations [Horia et. al. "Simple random search of static linear policies is competitive for reinforcement learning”] and [Islam et. al. “Reproducibility of benchmarked deep reinforcement learning tasks for continuous control”]. Please let us know any further references to add.
>
> 8. *“There is a distinction between self-supervised methods like BYOL and contrastive methods like SimCLR. The authors interchange them throughout the paper which is technically incorrect.”*
>
> Thanks for pointing this out, we have updated the text to only mention self-supervised learning.
>
> 9. *“CURL [...] seems to be a poor example to cite.”*
>
> We have removed the CURL citation in section 3.5.
>
> 10. *“Properly warm-starting / reusing features probably require lowering the learning rate for feature layers so that learning may not lose this initialization quickly after a few updates. Thus, it is unclear if feature learning experiments actually capture this or not.”*
>
> Thanks for this suggestion! We have added a baseline that uses warm starting and then a lower learning rate for the pre-trained features. This change does not affect performance much.
>
> 11. *“One of the main limitations of empirical study is that its conclusions are specifically tied to network architectures that use a tanh squashing function in the policy network.“*
>
> This is indeed a limitation, but we argue that this setting is still important. See general comment #1. We only consider networks with tanh, but some of our fixes apply to other networks.
>
> 13. *Regarding truly understanding why our fixes help. Especially for the critic network.*
>
> We state that “Our motivation has been to stabilize the actor output, but rapidly changing critic output might also be harmful and the critic network could similarly benefit from normalizing penultimate features.”. The failure modes of the critic are likely not as easy to interpret as the actor's failure modes. Please also see general comment #2.
>
>
> In light of our clarifications and completion of many requested experiments, we would like to ask if you are willing to reconsider your score, and also if there are any new concerns or additional questions we can respond to!

---

> > ### Comment · Reviewer_ti7w · 2021-11-20
> > **Discussion**
> >
> > Dear authors,
> >
> > Thanks for your response. I read the other reviews as well your response. Overall, I appreciate the additional experiments (especially the results on the additional tasks). One suggestion for authors is to make the paper updates using a colored text, so that it's easy to see what changes are made (rather than going through the tool). I'd like to request the following clarifications and reiterate concerns which weren't fully addressed: ​
> >
> > > "The failure modes of the critic are likely not as easy to interpret as the actor's failure modes .. see general comment #2"
> >
> > I think the `critic norm` is the least motivated part of the paper and gives me the impression of the paper being somewhat incomplete. Once the authors identified the problem, it seems they created a kitchen sink algorithm (combined) which works. I am not sure if citing the complexity of deep RL is a good enough reason to ignore trying to understand why something works. My worry is that if the proposed fixes gets adopted, this would likely make deep RL algorithms more mysterious instead of simplifying them.
> >
> > > "We only consider networks with tanh, but some of our fixes apply to "other networks."
> >
> > What kind of the other networks your fixes apply to. Can you provide empirical evidence for this belief? Otherwise, I feel it should be explicitly stated in the paper that the current fixes are only validated for networks with tanh.
> >
> > > "explaining variance in the sparse reward tasks"
> >
> > What is the reasoning behind including the combined++ agent in the appendix for fixing issues related to reward sparsity -- while we can add a bag of tricks to get higher performance on these tasks, how does that relate to the hypothesis presented in this paper?
> >
> > > "Over these 21 tasks, our combined agent improves average reward and decreases the average standard deviation by more than half."
> >
> > Average std/mean is somewhat  misleading here due to the large differences in std deviations (as you mentioned . For example, in Table 2, if I remove the walker task, the avg std of least variance method and baseline remains to be only ~40. That said, per-game results do indicate that the combined methods do seem somewhat encouraging. Same statement is true for avg mean scores across tasks.
> >
> > > "We have added an average score to the table which serves as an aggregate score. Robust metrics such as interquartile mean are unsuitable since we specifically are interested in outliers.
> >
> >
> > I agree with your point about being interested in outliers and as such, bottom percentiles seem more relevant. Since mean can be misleading as pointed above, I think it would make a lot more sense to report the entire distribution of scores (combined across all tasks and seeds).  Can you either plot the density function or even better the cumulative distribution function, such as [performance profiles](https://github.com/google-research/rliable#performance-profiles),  to see how well do the proposed approaches affect the outliers (both for Table 2 and for the overall performance).
> >
> >
> > Minor:
> >
> > > "updated the text to only mention self-supervised learning"
> >
> > You now cite SimCLR as a self-supervised method which is still incorrect.
> >
> > > "More detailed discussion about outlier runs in RL and discussing this in related work"
> >
> > This is somewhat cursorily addressed, it seems that you added citations without any *detailed* discussion.

---

> > > ### Author Response · Authors · 2021-11-21
> > > **Response, part 1**
> > >
> > > Thanks for your detailed comments and many constructive suggestions for how to improve the paper! We have updated our submission and responded to individual concerns below. Let us know of any further experiments to start or clarifications to make!
> > >
> > >
> > > 1. *One suggestion for authors is to make the paper updates using a colored text, so that it's easy to see what changes are made (rather than going through the tool).*
> > >
> > > This is a very good idea!  We have used it for this revision and will use it for future revisions too.
> > >
> > >
> > >
> > > 2. *I think it would make a lot more sense to report the entire distribution of scores (combined across all tasks and seeds). Can you either plot the density function or even better the cumulative distribution function, such as performance profiles*
> > >
> > > This is a great suggestion!  We have added a figure of performance profiles (Figure 15). The results still look good, especially for the worst-performing runs. Rliable seems to require that all tasks use the same number of seeds -- for this reason we break up performance profiles into the 5 original tasks and the 16 added tasks.
> > >
> > > 3. *I think the critic norm is the least motivated part of the paper*
> > >
> > > We agree that motivating and understanding the critic norm further would be good. We have added experiments regarding this to the appendix. Figure 19 shows that adding critic norm makes the Q-values much more stable during training, whereas they can oscillate aggressively otherwise. Oscillating Q-values likely impede learning for the actor. Please let us know if there are additional experiments you would like to see here!
> > >
> > > 4. *My worry is that if the proposed fixes gets adopted, this would likely make deep RL algorithms more mysterious instead of simplifying them.*
> > >
> > > This is a reasonable worry, we have added a sentence regarding this in the limitations section. However, we want to emphasize: 1) the ultimate goal of our paper is not to simplify RL agents. We want to show that one can significantly decrease the variance of SOTA agents without hurting the average reward. 2) This can be said of most papers that propose modifications to RL agents. 3) We do our best to motivate and provide intuition for our fixes. 4) Except for self-supervised learning (which has been proposed by multiple previous authors) the fixes we adopt are just a few lines of code.
> > >
> > > 5. *Once the authors identified the problem, it seems they created a kitchen sink algorithm (combined) which works.*
> > >
> > > We agree that the combined agent is a “kitchen sink” method in the sense that it uses multiple components. However, we want to highlight that we arrived at it in a principled way. We tested multiple methods and validated them independently. To achieve the best possible performance, we combined multiple orthogonal improvements that individually boosted performance.
> > >
> > >
> > > 6. *What kind of the other networks your fixes apply to. *
> > >
> > > Our fixes *apply* to other settings. E.g., normalizing penultimate features can be done for virtually any network. Similarly, disabling clipped double q-learning for the critic but not the actor does not rely on the tanh non-linearity. Whether our fixes *improve* performance is another question. See comments to reviewer 1 regarding trying additional agents.
> > >
> > > 7. *… Otherwise, I feel it should be explicitly stated in the paper that the current fixes are only validated for networks with tanh.*
> > >
> > > Stating that we focus on tanh networks explicitly is a good idea! We have stated this in the limitations section, but we believe that tanh networks are an important class.

---

> > > > ### Author Response · Authors · 2021-11-21
> > > > **Response, part 2**
> > > >
> > > > 8. *What is the reasoning behind including the combined++ agent in the appendix for fixing issues related to reward sparsity [...] how does that relate to the hypothesis presented in this paper?*
> > > >
> > > > The ultimate goal of our paper is to show that we can significantly decrease the variance of a SOTA agent without hurting the average reward. We believe that this is an important goal that is relevant for both research and applications of RL. After we finished experiments on additional tasks which were requested, it became clear that sparsity was a driver of residual variance. The results of the appendix show that this source of variance can also be attacked. This supports our claim that specifically optimizing for low variance can be fruitful and feasible.
> > > >
> > > > 9 *"we can add a bag of tricks to get higher performance on these tasks,"*
> > > >
> > > > We want to highlight that our modifications improve a recent SOTA agent, so these are not generic and widely-used “tricks”. Furthermore, simple modifications such as the one we propose can often have a significant impact -- examples include batch normalization and residual connections.
> > > >
> > > > 10. *You now cite SimCLR as a self-supervised method which is still incorrect.*
> > > >
> > > > Sorry about that! We thought SimCLR was self-supervised, e.g. the SimCLRv2 paper is titled “Big Self-Supervised Models are Strong Semi-Supervised Learners”. I now cite a survey paper instead.
> > > >
> > > > 11. *"More detailed discussion about outlier runs in RL and discussing this in related work"This is somewhat cursorily addressed, it seems that you added citations without any detailed discussion.*
> > > >
> > > > We would love to address this! However, we are not sure what kind of detailed discussion the reviewer is looking for. Our first revision modified the related work section to mention the contributions of many individual papers instead of citing them en-masse. In this revision, we expanded a little on two papers the reviewers specifically mentioned. Are there any specific papers we should discuss more in detail? If not, what changes would you like to see? There are some page limit constraints, so there’s no space to add 5 sentences per paper -- but we can add an extended background section to the appendix if needed.

---

> > > > > ### Comment · Reviewer_ti7w · 2021-11-21
> > > > > **Thanks & minor suggestions!**
> > > > >
> > > > > Thank you authors for the updates and your response. I am still not a fan of the combined++ agent but based on the overall discussion, I am now positive towards accepting the paper and would update my score to indicate this.
> > > > >
> > > > > > "Our fixes apply to other setting .. Whether our fixes improve performance is another question."
> > > > >
> > > > > This was confusing to me -- I thought when the paper said the fixes apply to other networks, I thought you meant they improve performance for other networks. Please add a clarifying statement explicitly stating that the fixes apply but whether or not they improve performance is left for future work.
> > > > >
> > > > > > "related work"
> > > > >
> > > > > I think an additional discussion in the appendix where you contextualize this work wrt prior work (which you currently cite) in more detail (what are similarities and differences and how are they connected) would be great.
> > > > >
> > > > > > "Very minor suggestion":
> > > > >
> > > > > If you can, please cite published versions of articles rather than the arxiv citation.

---

> > > > > > ### Author Response · Authors · 2021-11-22
> > > > > > **Thanks!!**
> > > > > >
> > > > > > Thanks for your feedback, many great suggestions, and appreciation of our work!!
> > > > > >
> > > > > >
> > > > > > 1. *I thought when the paper said the fixes apply to other networks, I thought you meant they improve performance for other networks. Please add a clarifying statement explicitly stating that the fixes apply but whether or not they improve performance is left for future work.*
> > > > > >
> > > > > > We have added such a statement to the limitations section. Thanks for highlighting this, our initial wording was definitely confusing!
> > > > > >
> > > > > > 2. *I think an additional discussion in the appendix where you contextualize this work wrt prior work (which you currently cite) in more detail (what are similarities and differences and how are they connected) would be great.*
> > > > > >
> > > > > > Thanks for this suggestion, we will add an appendix for extended related work discussion. Therein we will dedicate at least a couple of sentences to each paper we cite in section 5. We will likely not be able to finish writing this appendix tonight but will have it done within a few days. After that, we will update the manuscript as soon as possible.
> > > > > >
> > > > > > 3. *If you can, please cite published versions of articles rather than the arxiv citation.*
> > > > > >
> > > > > > Thanks, this is a good suggestion! We have exchanged arXiv references for conference references in all cases where we’ve found a corresponding conference publication. The number of arXiv references has decreased from 36 to 8. These changes are not highlighted in red, we don’t know how to achieve this with BibTeX.

---

### Official Review · Reviewer_cHfn · 2021-11-01

**Correctness:** 3
**Technical Novelty And Significance:** 3
**Empirical Novelty And Significance:** 3
**Recommendation:** 10
**Confidence:** 4

**Main Review:**

Strength:
The main strength of the paper is its focus on an important problem and studying it effectively. Being able to pinpoint the problem of variance to exploding activation of actions is definitely a success. Various experiments are conducted based on well-formed hypotheses trying to rule out other suspected sources.

The main weakness is not thoroughly discussing the result in table 2. Which approaches really worked? It is claimed that “Normalizing activations seems to stably improve performance and decrease rewards”. If I look at it closely, actor normalization achieves the second largest variance on finger turn. Actor+critic normalization does not achieve a low variance on that task either.

Another weakness is that the choice for combining different strategies isn’t justified. The penalty got mixed results from table 2 and so did the layer norm. Early contrastive learning, which is added in the combined fix, came out of nowhere. To me, it seems like different combinations were tried and this one somehow worked. Can you shed some light on it? How does Figure 4 look like when we only try normalization or penalty or a combination of just those two? What would the mean and the standard deviation for the combination of just these two be?



**Summary Of The Paper:**

This paper is an empirical study of the issue of high variance in RL. Learning continuous control from pixels is adopted as the setup. The authors demonstrated that the failure of a few runs is responsible for much of the variance. To look for the source of the variance, they separately studied the impact of the initial and training-time randomization as well as variation in representation learning between runs, none of which were found to be majorly responsible for the variance. The authors hypothesized and isolated the exploding activation of the actions to be the main source of the variance, which cause some runs to saturate beyond salvation and fail at performing. The hypothesis is tested by employing several measures to counter this phenomenon. Employing normalization of the penultimate features turned out to be particularly helpful.


**Summary Of The Review:**

It is a paper on an empirical study of an important question in RL with satisfactory claims and fixes. In the latter part of the paper, some thorough discussions are missing, and the choices made around the combined fixes are not well motivated, for which I would like to hear from the authors.

*** Updated ***

The authors addressed all my concerns satisfactorily. They even addressed a last-minute question about a highly-related work that I didn't notice before. The authors clearly described the difference between that work and theirs.

More elaborately, the authors promptly and elaborately discussed a comparison to this newly discovered work. They not only showed that they can discuss this work in their submission fairly, but they can also utilize it to improve their submission. Moreover, their new results showed that the existing work was inferior to their proposed solution. The authors' explanation also made sense that the existing approach normalized the pre-tanh action, which limited the action space too restrictively.

Hence, I increased my score from 6 to 10. Thank you!

---

> ### Author Response · Authors · 2021-11-18
> **Review 2**
>
> Thanks for your encouraging review and suggestions for improving the clarity of the paper! We address individual points below:
>
>
> 1. *“The main weakness is not thoroughly discussing the result in table 2. Which approaches really worked?”*
>
> We have added a column with average performance and expanded the discussion. The activation normalization performs the best by this average metric -- the two versions have the two highest rewards and the two lowest variances. We have clarified that we refer to average performance. Furthermore, the penalty improves the performance for this metric too.
>
> 2. *“Another weakness is that the choice for combining different strategies isn’t justified. [...] Can you shed some light on it?”*
>
> We hope that the added column with averaged performance clarifies this. The activation normalization performs the best by this average metric. Layer-normalization sometimes performs poorly, so we don’t use it. The penalty method improves average reward and decreases average standard deviation. Contrastive/self-supervised learning also slightly improves average scores on the five tasks we tried, see Table 1. The combined agent was obtained by selecting orthogonal modifications that individually perform well, we did not try multiple combinations.
>
> 3. *How does Figure 4 look like when we only try normalization or penalty or a combination of just those two? What would the mean and the standard deviation for the combination of just these two be?*
>
> We have added the requested plots for the normalization and penalty agent in Figures 13 and 14. Trying this combination of them is a good idea which we have added to our to-do list. However, we’d want to emphasize that testing all combinations of all methods is computationally infeasible for us. Our main contribution is in finding one good combination that significantly decreases variance.
>
> We have clarified the results in table 2, which was identified as a major weakness. We have furthermore shown how the modifications of table 2 transfer to additional tasks. In light of this, we would like to ask if there are any experiments we can run to make you consider increasing your score, and also if there are any new concerns or additional questions we can respond to!

---

> > ### Comment · Reviewer_cHfn · 2021-11-24
> > **Weaknesses addressed**
> >
> > Thank you for your response. It clarifies the major concerns.

---

> > ### Comment · Reviewer_cHfn · 2021-11-26
> > **A highly related work**
> >
> > Thanks again for the response.
> >
> > It came to my attention that the below-published work addresses the same action saturation issue with a similarly simple solution.
> >
> > Your paper stumbles on this issue by pursuing the high-variance problem ensuing from failed runs. But could we not just use this existing work to address the high-variance problem as well? Moreover, no reference, comparison, or discussion is included in your work, which seems necessary to accept yours as a scholarly work.
> >
> > Wang, C., Wu, Y., Vuong, Q., Ross, K. (2020). Striving for simplicity and performance in off-policy DRL: Output normalization and non-uniform sampling. In International Conference on Machine Learning

---

> > > ### Author Response · Authors · 2021-11-26
> > > **Thanks for this pointer, we will cite and discuss!**
> > >
> > > Thanks for bringing this paper to our attention, we were not aware of it. We apologize for missing it in our literature review!
> > >
> > > We will update the paper to cite Wang et. al. and mention our remarks given below. Specifically, we will cite the paper in sections 1, 3.6, and 5 and clarify that saturating actions have been observed when removing the entropy of a SAC agent. We will also cite the paper in section 4 and to see if it decreases variance we will add the proposed scheme to Table 2. We have started this experiment. Please let us know of any other places where we should discuss it!
> > >
> > > We want to highlight a few points in how our work differs from Wang et. al. 1) Firstly, we want to emphasize that the main conclusion of our work is orthogonal to Wang et. al.: we demonstrate that it is possible to dramatically decrease variance for a SOTA agent without hurting average performance. 2) Furthermore, we show that many possible issues (initial seeds, environment sensitivity, etc.) are not drivers of variance. 3) We also consider fixes that are completely independent of the problems normalization solves and show that these decrease the variance significantly, see sections 4.2 and Appendix C. 4) Wang el. al dedicates substantial space (section 5) to a new strategy of sampling the replay buffer. This is completely orthogonal to our paper. Regarding overlap, see below:
> > >
> > > Regarding saturating action distributions. Wang et. al. studies this issue in a rather artificial setting -- by removing the entropy bonus of a SAC agent. The entropy bonus is integral to SAC, so this is not a very “realistic” setting and it is not clear from the experiments of Wang et. al. if saturating tanh is an issue “in practice”. In contrast, we study a SOTA agent where this issue arises naturally. Furthermore, Wang states that “the principal contribution of the entropy term in the SAC objective is to resolve the squashing exploration problem”. The squashing exploration problem is simply the fact that for large $|\mu|$, $\tanh( \mu + \epsilon)$ will vary very little with $\epsilon$, which effectively eliminates the exploration. For DDPG, which we consider, the noise is added outside the tanh operator, so there is no squashing exploration problem. In contrast, we show that vanishing gradients is the main issue caused by saturating tanh. Vanishing gradients and insufficient exploration are two very different problems.
> > >
> > > Regarding normalization. First, we note that the proposed normalization scheme is mathematically distinct from ours and that we do not normalize the output directly. We already cite [1,2] which similarly proposes normalization strategies in RL, and we do not claim that we are the only ones to consider normalization in RL. Furthermore, Wang et. al. only appears to study if normalization helps for the actor. In contrast, we demonstrate that using it for the critic can improve the performance dramatically (see Table 2), and provide experiments demonstrating why (See Appendix D). We will highlight these experiments more.
> > >
> > > [1] Weight Normalization: A Simple Reparameterization to Accelerate Training of Deep Neural Networks (Salimans and Kingma, 2016)
> > >
> > > [2] Spectral Normalisation for Deep Reinforcement Learning: An Optimisation Perspective (Gogianu et al 2021)

---

> > > > ### Comment · Reviewer_cHfn · 2021-11-27
> > > > **reply**
> > > >
> > > > Thanks for the quick response.
> > > >
> > > > The clarification was helpful. It is true that there is more to your work than action saturation. Let's consider this. Are the additional insights significant enough so that if we drop the action saturation issue, the work still remains strong?
> > > >
> > > > My understanding is that you are showing that action saturation is the main contributing factor to high variance. Then is it possible that Wang et al.'s solution fully addresses this issue even though they introduced it for a related but different issue, that is, squashing exploration? If so, is the paper still interesting? If so, please do keep the option of describing that narrative open. It will only improve the work and the overall knowledge of the community.

---

> > > > > ### Comment · Reviewer_ti7w · 2021-11-28
> > > > > **agreed with reviewer comment**
> > > > >
> > > > > Based on this comment, I feel this discussion of this very related work seems quite important. I'm also updating my score to 6 to reflect that this new information should have been properly addressed in the original submission and the paper might benefit from another round of reviews.

---

> > > > > ### Author Response · Authors · 2021-11-29
> > > > > **reply, part 1**
> > > > >
> > > > > Thanks for your comments and for engaging with us! Below we answer individual points and commit to additional and specific updates to the paper. Since it is past the paper update deadline, we unfortunately cannot implement these changes in openreview at this time. Please let us know of any further comments or clarifications we can make, and if there are aspects of Wang et. al. which we should discuss more!
> > > > >
> > > > > 1. *is it possible that Wang et al.'s solution fully addresses [action saturation] even though they introduced it for a related but different issue, that is, squashing exploration?*
> > > > >
> > > > > This is a good question! Our experiments for Wang et. al. are finished and will be added to Table 2. For now, they are given below:
> > > > >
> > > > > | metric      | acrobot  | finger turn  | hopper hop | reacher  | walker  | avg.  |
> > > > > |------------|-----------------|------------------|------------|--------------|-------------|-------|
> > > > > | $\mu$        |  13.3           |  306.3           |  216.9     |  699.4       |  952.1      | 437.6 |
> > > > > | $\sigma$    |  16.6           |  190.4           |  21.4      |  180.4       |  17.2       | 85.2  |
> > > > >
> > > > > The method receives roughly the same low variance as actor norm but achieves lower average rewards. Importantly, it seems completely unable to learn a good policy for the acrobot task, which lowers the variance somewhat artificially. We hypothesize that this environment requires |a| \approx 1, which becomes impossible if the pre-tanh activations are directly constrained as Wang et. al. propose. E.g., for a one-dimensional action space, the proposal of Wang et. al. changes the action space from [-1,1] to [tanh(-1), tanh(1)] (modulo a noise term, which is often set to 0 during evaluation). Furthermore, the method of Wang et. al. performs poorly compared to both norm and the combined agents. Thus, their method does not address all problems that we address and performs worse than our actor normalization scheme.
> > > > >
> > > > >
> > > > > 2. *Are the additional insights significant enough so that if we drop the action saturation issue, the work still remains strong?*
> > > > >
> > > > > Our main contribution, we believe, is showing that we can significantly decrease variance (the average standard deviation across 21 tasks decreases by >3x) for a SOTA agent, without hurting average performance. In empirical RL, performance is typically measured by average reward. We show that optimizing directly for lower variance is fruitful and feasible. Lower variance can be more important than average reward for sensitive applications, and we believe that focusing on low-variance agents could be an important direction for the RL community. Our low-variance results are completely orthogonal to Wang et. al.
> > > > >
> > > > > Furthermore, the normalization of the critic and optimizations for sparse tasks are crucial components of the final agent. These methods are of course unrelated to saturating tanh and improve performance individually. We also provide experiments showing that some issues (e.g. initial seed and environment sensitivity) are not especially relevant for variance -- these results can inform future RL researchers interested in variance.
> > > > >
> > > > > Finally, we want to highlight that the saturating action issue we describe, loss of gradients, is distinct and different from the lack of exploration in SAC without entropy (which is a rather artificial setting). The normalization methods we consider are also different, and importantly we do not directly constrain the action space. So even given this prior work on saturation, we believe that our results on saturating tanh can be useful additions to the literature.

---

> > > > > > ### Author Response · Authors · 2021-11-29
> > > > > > **reply, part 2**
> > > > > >
> > > > > > 3. *My understanding is that you are showing that action saturation is the main contributing factor to high variance.*
> > > > > >
> > > > > > We’d like to emphasize a nuance here. For the walker task, which we illustrate in Figure 3, saturating actions are likely the main driver of variance. However, this is one out of 21 tasks, and we have tried to avoid making such a statement broadly. Instead, we say that saturating tanh is *one* important source of variance, e.g. in the abstract, we write “ We show that one cause for outliers is numerical instability which leads to saturating nonlinearities”. Saturating tanh is certainly not the only issue -- multiple other fixes which are not related to saturating action distributions also improve performance. Perhaps this nuance has not been stated clearly enough, and we can clarify further.
> > > > > >
> > > > > > 4. *is the paper still interesting? If so, please do keep the option of describing that narrative open. It will only improve the work and the overall knowledge of the community.*
> > > > > >
> > > > > > Thanks for this point! We believe the paper still has significant merits, but think that it is reasonable to further highlight aspects of the paper that are orthogonal to saturating tanh. Specifically, a smaller version of Figure 19 can be put in the main paper to demonstrate how the critic suffers from instability. Furthermore, we can expand on the fixes considered in Appendix C in section 4.2 the main paper. To make space we can 1) move section 4.3, which details learning rate stability, to the appendix. 2) Remove row 2 of Figure 3, which isn’t crucially needed for our argument. Saturating tanh is the focus of page 6 and the bottom of page 5, which represents only ~12% of the space of the paper. By removing row 2 of Figure 3 (and text associated with row 2), it should be >= 10% of the paper.
> > > > > >
> > > > > > As for the narrative, we will focus more on our empirical results of decreasing the variance for a SOTA agent and discuss how the RL community might benefit from low-variance agents.  As stated in earlier comments, we will of course cite and discuss Wang et. al. and will clarify that saturating tanh is just one cause of variance which we address. It is past the paper update deadline, so we cannot implement these changes in openreview immediately. If you have any suggestions regarding the narrative, please let us know!

---

> > > > > > > ### Comment · Reviewer_cHfn · 2021-11-29
> > > > > > > **This resolves all my concerns**
> > > > > > >
> > > > > > > Thank you for the response. It was a last-minute discovery and your responses assured me that you can discuss this paper fairly and utilize that to increase the quality of the paper significantly. Hence, I am increasing my score.

---

### Official Review · Reviewer_N91S · 2021-11-02

**Correctness:** 4
**Technical Novelty And Significance:** 3
**Empirical Novelty And Significance:** 3
**Recommendation:** 6
**Confidence:** 4

**Main Review:**

As pointed out in the paper, variance in RL is one of the main challenges to efficient and relevant research today, and this paper is a well-written exploration of a possible cause of this variance from the DRQv2 algorithm.
The paper studies the performance of the DRQv2 algorithm on several continuous control tasks, arguing that the variance in RL performance over several runs is primarily driven from a bimodal distribution of training runs.
Some of the runs learn well, but some get stuck early on and don't learn. Several possible reasons are proposed for this, including poor initialization, feature learning, and unforgiving environments. The authors then argue that saturating tanh nonlinearities are to blame. Two remedies are proposed: a penalty on the squared norm of the action, and a capacity reduction of the model where the features of the penultimate layer are normalized and a simple linear layer is used to produce the action.

Overall I think the paper identifies an interesting area and an interesting idea, but there is currently a lack of rigour in a lot of the arguments and seems to have some incorrect statements. Furthermore, there is a lack of references to some existing work with quite similar approaches. These points are elaborated below

Comments:

1. In equation 2 and 3, it is claimed that the action $a$ is produced as $a_\theta(s) = \tanh (\mu_\theta(s) + \epsilon \cdot \sigma_t), \epsilon \sim {\cal N}(0, 1)$. However, in DDPG, the action is generated from a clipped truncated normal which is centered at $\mu = \tanh(f(s))$ for some learned \(f\) mapping states to $\mathbb{R}$. Indeed, in the authors' code this seems to be exactly the procedure followed in the `Actor` method in the drqv2.py file, unlike as described in eqs 2, 3. It's possible I may have misunderstood this, so could the authors comment on this discrepancy?

2. The authors claim that the failure to learn is primarily due to runs that get 'stuck' early on, with figure 1 as evidence. However, it's quite hard to read figure 1 due to the large number of overlapping lines making it hard to tell the relative number of runs. Could the authors add a third row to this figure showing a histogram of the final return achieved by each run? The authors' claim can then be reinforced if the histogram is bimodal with many at zero return and the 'good' return. Continuing on figure 1, it is unclear to me how well this supports the argument that the final returns are bimodal, in particular with reference to the `hopper` environment where it appears the majority of the runs are still improving after 1,000 episodes. If we run for longer, do they all continue to improve?

3. Section 4 argues that the cause of the stuck runs is saturation with the tanh nonlinearity. However, I think this section seems to leave out some important details which could help show convincingly that the 'stuck run' cause is the tanhs. Firstly, it would be useful to have a version of fig 3 that is done on a population level, like figure 1. This would allow the reader to compare the runs in figure 1 and 3, and verify that this behaviour happens consistently. (It may be too busy to show these all runs, but e.g. a large high-resolution plot showing 20 randomly-selected runs from the dataset of 40 seeds would be a great figure to eventually have in an appendix). Secondly, it would be useful to focus on the size and gradient of the final layer specifically. As it is, it's hard to judge what's happening when looking at the norm of l and w since they are taken over the whole actor's weights. Plotting the pre-tanh $a_\theta^{\text pre}$ would also be more useful than a flat saturated 1 value for the norm of the action. A logarithmic scale would also be useful for the gradient norm.

4. In section 4, did the authors try initializing the last layer's weights to be much smaller than the other layers? It has been suggested previously that the last policy layer should be initialized to 100x the weight of the other layers [1], and it seems that this may combat the saturating nonlinearities.

5. It is surprising to me that the $\lambda$ value 1e-6 is sufficient, as this seems very small. Could the authors elaborate on why such a small value is sufficient to change the training dramatically?

6. A similar idea, that of normalizing the features extracted for the agent in an off-policy algorithm, has been explored in quite a bit of detail for the DQN. In particular, [2] investigates weight normalization for DQN, and in [3] spectral normalization is applied in the layer before the action outputs, which seems quite similar in spirit to the normalization in this paper. [3] Also shows that this increases robustness with respect to hyperparameters. Can the authors comment on this work?

7. At the moment the technique presented is only evaluated on DDPG. However, it seems that this saturating tanh problem could be present any other RL algorithm with bounded action range. If this method could be shown to consistently help with other algorithms, this would significantly increase the significance of this result. I appreciate that computation resources are not free, but if the authors could show even preliminary results that this method works with e.g. SAC, PPO, etc, this would be very significant.

I am happy to reconsider my recommendation based on feedback from the authors and other reviewers.

Summarizing:

Strengths:
+ Convincing argument about the importance of the tanh saturating in DDPG
+ Thorough evaluation of a few different ways of addressing this problem

Weaknesses:
+ Missing some relevant related work on normalizing image features in off-policy RL
+ Some possible incorrect statements about the action distribution
+ Plots that could be made easier to read and interpret
+ Limited experimental evaluation only on DDPG


[1] What Matters for On-Policy Deep Actor-Critic Methods? A Large-Scale Study (Andrychowicz et al, ICLR 2021)

[2] Weight Normalization: A Simple Reparameterization to Accelerate Training of Deep Neural Networks (Salimans and Kingma, 2016)

[3] Spectral Normalisation for Deep Reinforcement Learning: An Optimisation Perspective (Gogianu et al 2021)

----------------------------------------------------------
Update 2021-11-21
Based on the authors' additional results, correction of some inaccuracies and comparison to previous work on normalization, I have raised my score to 6.

**Summary Of The Paper:**

The paper studies the variance of the DDPG algorithm on continuous control tasks. They identify that the variance in rewards arises in part due to the failure of some runs to learn. They argue that the failure to learn is due to saturating tanh nonlinearities, and suggest two methods to address this problem.

**Summary Of The Review:**

The paper identifies an interesting phenomenon and has good proposed fixes, but there is currently a lack of rigour/convincing arguments and seems to have some incorrect statements. There is a lack of references to some existing work with similar approaches. The method has somewhat limited significance as it is only demonstrated to be useful with DDPG.

---

> ### Author Response · Authors · 2021-11-18
> **Review 1**
>
> Thanks for your detailed review and fruitful comments. We address individual points below:
>
>
> 1. *Questions regarding the policy parametrization.*
>
> The noise in eq (2,3) should be outside of the tanh operator, thanks for catching this! We have fixed this. The policy of the DRQv2 agent does not use truncated normal noise. The code uses dist.sample(clip=None) for generating the policy, and clip=None implies no truncation. Truncation is only used for training via the function call dist.sample(clip=self.stddev_clip). Different DDPG implementations/agents use different noises such as normal noise [1] or noise from an Ornstein-Uhlenbeck process [2]. We have expanded the background to mention that the noise type is essentially an implementation choice.
>
> [1] Spinning up RL, (​​https://github.com/openai/spinningup/blob/038665d62d569055401d91856abb287263096178/spinup/algos/pytorch/ddpg/ddpg.py#L228).
>
> [2] Lillicrap et. al. Continuous control with deep reinforcement learning.
>
>
> 2. *Weakness: Limited experimental evaluation only on DDPG*
>
> We hope that adding further tasks has made our experimental evaluation less limited. Furthermore, see general comments #2.
>
>
> 3. *“it would be useful to have a version of fig 3 that is done on a population level”*
>
> We have added a figure of all 40 runs to the appendix as Figure 15. The correlation between saturated tanh and poor rewards is consistent.
>
> 4. *“ A logarithmic scale would also be useful for the gradient norm”*
>
> We have plotted this quantity in Figure 15 in the Appendix.
>
> 5. *“It is surprising to me that the value 1e-6 is sufficient, as this seems very small. Could the authors elaborate on why such a small value is sufficient to change the training dramatically?”*
>
> This is a great point! As shown in Figure 15, the actor gradients can sometimes reach near-zero when the tanh non-linearity is saturated. If we take the gradients to be zero, only the action penalty contributes to the loss term. The adam optimizer, which we use, is invariant under multiplying the loss function by any non-zero constant (modulo a small epsilon which is 1e-8 as per PyTorch default). Thus, in this regime, the numerical value of $\lambda$ does not matter much -- both $\lambda=100$ and $\lambda=1-6$ would yield a similar result. We will clarify this in the main text.
>
> 6. *“Can the authors comment on [spectral normalization of Gogianu et. al. and weight normalization of Salimans et. al]?”*
>
> Thanks for these pointers! We have cited both papers and have added spectral normalization following [3] as a baseline in Table 2. As can be seen, it is not particularly effective. We will add weight normalization to our todo-list. Since the paper is relatively old and well-known, and since the method isn’t widely used in RL, we suspect that the results might not be very good.
>
> 7. *“In section 4, did the authors try initializing the last layer's weights to be much smaller than the other layers? It has been suggested previously that the last policy layer should be initialized to 100x the weight of the other layers [1], and it seems that this may combat the saturating nonlinearities.”*
>
> Thanks for this suggestion! We have added this strategy as a baseline in Table 2. As can be seen, it does not work very well.
>
> 8. *Request for two additional metrics: gradient of final layer and pre-tanh activations.*
>
> These quantities were not logged during our initial runs, so plotting them would require rerunning the agent. We have put this on our experimental “to-do” list.
>
> 9. *Comments regarding whether the tasks have bimodal returns or not.*
>
> Thanks for this point, we have reformulated some sentences to clarify our claims. We do not claim that all tasks will have a bimodal distribution, but that outliers are important and often drive variance. E.g. In Figure 1 we originally stated that “The high variance is often caused by a few outlier runs that completely fail to learn. This is especially clear in the walker task.”. The term “often” is perhaps not suitable here, since we only consider 5 tasks. We have rephrased the abstract and intro to state that “outlier runs is an important source of variance” rather than them “often” dominating variance. We have also rephrased the caption to Figure 1 to remove the word “often” and instead state that “For some tasks, the high variance is dominated by a few outlier runs that completely fail to learn.” We apologize that our writing was sloppy in this regard, and hope that the reviewer approves of this fx. The updated paper focuses more on quantitative improvements.
>
> In light of our clarifications and completion of many requested experiments, we would like to ask if you are willing to reconsider your score, and also if there are any new concerns or additional questions we can respond to!

---

### Author Response · Authors · 2021-11-18
**Rebuttal**


We thank the reviewers for their thoughtful reviews and great feedback. We are especially happy that the reviewers believe that the area we study is important (cHfn, J8Mk) and interesting (ti7w, N91S). The reviews are filled with good ideas for improvements, and we have added many of the requested experiments. The major updates in this version are:

- We add several requested baselines: Smaller weights for the final layer, a smaller learning rate for pre-trained weights, and spectral normalization following [1]. These extra baselines do not beat our methods.
- Following requests for more tasks, we have added experiments on 16 additional tasks. Our results generalize well to additional tasks, we can decrease the average standard deviation by a factor 2-3 across the 21 tasks. These additional tasks show that once stability is addressed, sparsity is a major cause of residual variance, and addressing it can give further improvements. We’d like to emphasize that improvements such as these could have dramatic effects on the deployment of RL agents in the real world.

We have not been able to complete all requested experiments, but hope to add more. We would also like to state that we will release code to reproduce our experiments upon publication, even though our contributions are mainly implementation details built upon the agent of [2]. We apologize that completing the rebuttal took a long time, we are happy to interact with the reviewers until the deadline! Below we address two general points brought up in the reviews. Other comments are given in response to individual reviews.

# General comment 1

One common concern is the lack of generality of our methods. This is indeed a limitation of our work, and one which we identify. However, many import RL papers similarly show improvements on a single setting, e.g. [3,4,5]. We would also argue that our setting is important for several reasons: 1) continuous control for pixels is an important domain with real-world applications. 2) The dm-control suite is an important and well-studied benchmark. 3) We use a state-of-the-art agent (DRQv2 [2]) and demonstrate that we can substantially decrease its variance. 4) Tanh rectification is a common and natural non-linearity for bounded action spaces, these are common since e.g. robotic arms cannot exert arbitrary forces.

# General comment 2

Another common question is whether we “really” understand what is going on. We are grateful for this point and have further emphasized the epistemological limits of our work. Due to the complexity of modern deep reinforcement learning, pinpointing sources of variance with mathematical rigor is beyond our reach. Instead, our arguments and analyses are empirical and are intended as such. Our most important result is quantitative -- we show that we can decrease the standard deviation significantly across 21 tasks for a SOTA agent. In addition to this quantitative contribution, we believe that it is worthwhile to provide intuition and motivation for our modifications.


### References

[1][Gogianu et. al. Spectral Normalisation for Deep Reinforcement Learning: an Optimisation Perspective]

[2][Yarats et. al. Mastering Visual Continuous Control: Improved Data-Augmented Reinforcement Learning]

[3][Hessel et. al. Rainbow: Combining Improvements in Deep Reinforcement Learning]

[4][Hafner et. al. Learning Latent Dynamics for Planning from Pixels]

[5][Guez et. al. Deep reinforcement learning with double q-learning]

---

### Decision · Program_Chairs · 2022-01-20

**Decision:**

Accept (Poster)

**Comment:**

The reviewers unanimously appreciated the quality of the experiments. The main point raised was about the related work by Wang et al. but that was addressed by the authors in the rebuttal. I thus encourage the authors to make sure that discussion is reflected in the final version of their work.